# Integrated multi-omics analysis of the microbial profile characteristics associated with pulmonary arterial hypertension in congenital heart disease

Jiahui Xie,[1] Xiaoyu Zhang,[2] Liming Cheng,[3] Yao Deng,[1] Haobo Ren,[1] Minghua Mu,[1] Liang Zhao,[1] Chunjie Mu,[1] Jiaxiang Chen,[1] Kai Liu,[4] Runwei Ma[1]

**ABSTRACT**   Dysregulation of immune and inflammatory cells around blood vessels and metabolic dysfunction are key mechanisms in the development of pulmonary arterial hypertension (PAH). The homeostasis of the human microbiome plays a crucial role in regulating immune responses and the progression of diseases. For pulmonary arterial hypertension associated with congenital heart disease involving body-lung shunt (PAH-CHD), the potential impact of the microbiome on the "gut-lung axis" remains underexplored. This study recruited 15 healthy individuals and 15 patients with pulmonary arterial hypertension due to congenital heart disease from Fuwai Yunnan Hospital, Chinese Academy of Medical Sciences, and Kunming Children's Hospital. We performed differential analyses of metabolites and microbiota from both the gut and lower respiratory tract for these two groups. The goal was to investigate the "gut-lung axis" microbiome and metabolome profiles in children with PAH-CHD and to analyze the interrelationships between these profiles. Ultimately, we aim to propose the potential value of these profiles in aiding diagnosis. The results indicated that the gut and pulmonary microbiota of children with PAH-CHD are characterized by an increased abundance of beneficial symbionts, which are closely linked to changes in the metabolome. Metabolite functional enrichment analysis revealed energy metabolism reprogramming in the PAH-CHD group, with active metabolic pathways associated with bile acid secretion and carnitine homeostasis. Moreover, the differential expression of metabolites was correlated with right heart function and growth development.

**IMPORTANCE**   Previous studies have primarily focused on the relationship between the gut microbiome and PAH. However, the impact of microbial homeostasis on the progression of PAH-CHD from the perspective of the gut-lung axis has not been adequately elucidated. Our study utilizes an integrated multi-omics approach to report on the differential characteristics of gut and lung microbiota between children with PAH-CHD and reference subjects. We found that microbiota influence the pathological changes and disease manifestations of PAH-CHD through their metabolic activity. Additionally, alterations in metabolites impact the microbial ecological structure. Our findings suggest that modulating the microbiome composition may have positive implications for maintaining and regulating the immune environment and pathological progression of PAH-CHD.

**KEYWORDS**   human microbiota, metabolome, PAH-CHD, inflammation, reprogramming of energy metabolism, growth failure

P ulmonary arterial hypertension (PAH) is a form of precapillary pulmonary hypertension (PH) that results from the remodeling of the small pulmonary arteries, leading

**Peer Reviewer** Hawraa Natiq Kabroot AL-Fatlawy, University of Kufa College of Medicine, Najaf, Iraq

Address correspondence to Kai Liu, ynkmlk@foxmail.com, or Runwei Ma, marw0102@163.com.

Jiahui Xie, Xiaoyu Zhang, and Liming Cheng contributed equally to this article. The order of authors is based on their contributions.

The authors declare no conflict of interest.

See the funding table on p. 17.

to an increase in pulmonary artery pressure. PH is diagnosed when the mean pulmonary artery pressure (mPAP), measured by right heart catheterization, is greater than 20 mmHg at sea level in the first 3 months of life (1, 2). Congenital heart disease (CHD) with body-lung shunts is a major cause of PAH in children, with the age of onset depending on the nature of the defect (3). Children with congenital heart disease exhibit congenital anomalies of the cardiovascular system, particularly left-to-right shunting, which results in a continuous volume and pressure overload of the pulmonary circulation. This gradual change from reversible to irreversible pulmonary vascular lesions increases the risk of pulmonary complications (4). It has been demonstrated that early clarification of the extent and nature of pulmonary vasculopathy, in conjunction with aggressive surgical correction of intracardiac malformations, has a positive clinical outcome in children with PAH-CHD.

Chronic inflammation and immune imbalance lead to extracellular matrix remodeling and fibrosis, resulting in decreased pulmonary vascular compliance and ultimately causing PAH (5, 6). Based on this, microbial homeostasis has received significant attention. Kim et al. (7) suggested that variations in gut microbiota may effectively predict PAH (7). Gut microbiota influence the host's immune response, metabolic processes, and resistance to pathogens through their collective metabolic activities and interactions, thereby impacting the onset and progression of diseases (8, 9). The specific anatomical and hemodynamic conditions of CHD may alter the local tissue microenvironment and influence the ecological structure of the microbiota, which is further involved in the formation and development of PAH. Previous studies have shown that children with PAH-CHD exhibit changes in pulmonary microbiota and metabolites, with a decrease in *Bacteroidetes* and an increase in *Lactobacillus*, *Alicycliphilus*, and *Parapusillimonas*. Disruptions in purine, glycerophospholipid, and pyrimidine metabolism further suggest the involvement of pulmonary microbiota in PAH (10). Nevertheless, there is still a paucity of microbial community mapping for specific alterations in the gut-lung axis in the recognition of PAH-CHD. Furthermore, the interactions between microbial functional properties, metabolic activities, and the host require further investigation.

This study performed 16S rRNA sequencing, as well as macroeconomics and metabolomics analyses, on bronchoalveolar lavage fluid (BALF), fecal, and blood samples from children with PAH-CHD and healthy controls. The aim was to characterize the gut and pulmonary microbiota of PAH-CHD patients and to explore their association with the "gut-lung axis" in PAH-CHD. Additionally, we conducted a joint analysis of the correlation between cardiac ultrasound parameters and metabolomic profiles to obtain a biological understanding of the clinical features.

## RESULTS

### Baseline characteristics of participants

This study included 15 children diagnosed with PAH-CHD and 15 healthy control children matched for age and gender. Table S1 summarizes the clinical characteristics of the participants, including demographic information, PAH etiology, and echocardiographic variables.

### Changes in the microbiota composition of PAH-CHD children

First, the composition and diversity of the gut and lung microbiota were analyzed. Alpha diversity analysis assessed the richness and homogeneity of species diversity within each cohort. The gut microbiota exhibited higher richness and diversity in the PAH-CHD group (pielou_e, $P = 0.04$; Shannon, $P = 0.04$), while the pielou_e ($P = 0.97$) and Shannon ($P = 0.90$) indices of lung microbiota showed no significant differences between groups in terms of microbial diversity (Fig. 1A; Fig. S1A). Furthermore, principal coordinate analysis (PCoA) based on Bray-Curtis dissimilarity demonstrated no significant differences in the composition of gut and lung microbiota between groups (Fig. S1B). By plotting a circos diagram to illustrate the relationship between grouping and species

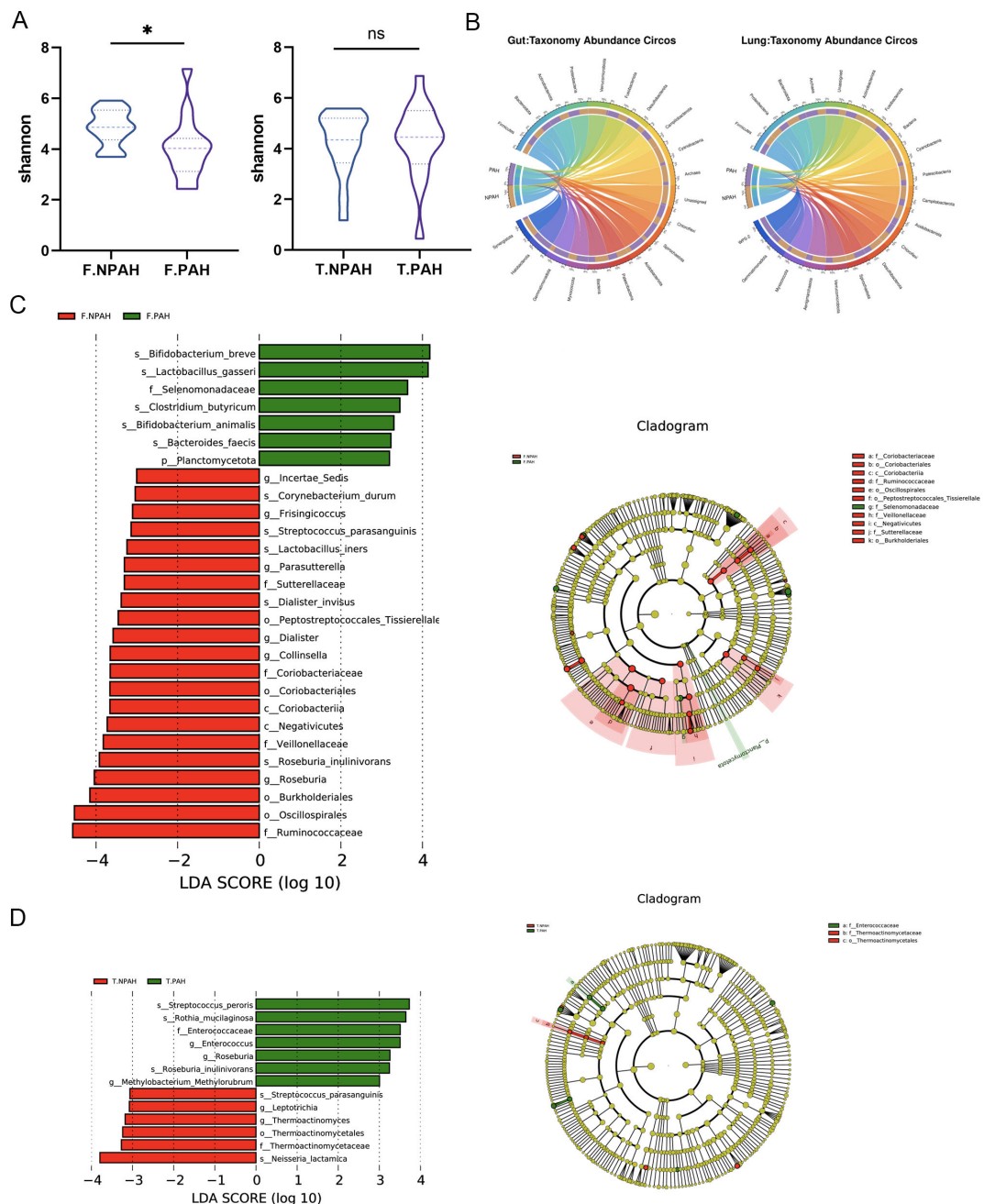

**FIG 1** Characteristics of the microbiota in children with PAH-CHD and healthy controls. (A) Comparison of gut and lung bacterial α-diversity (Shannon) between PAH-CHD and healthy control groups (Student's *t*-test). (B) Circos plot showing the top 20 microbial abundances in the PAH-CHD and healthy control groups. (C) LEfSe analysis reveals significant differences in the enrichment of gut bacterial taxa between the PAH-CHD group and healthy controls, along with their phylogenetic tree Linear Discriminant Analysis (LDA > 3). (D) LEfSe analysis reveals significant differences in the enrichment of lung bacterial taxa between the PAH-CHD group and healthy controls, along with their phylogenetic tree (LDA > 3). PAH: PAH-CHD groups; NPAH: healthy controls; F-: derived from fecal samples; T-: derived from BALF samples.

abundance, we found that while the dominant microbial taxa were similar between groups, their intra-group compositions differed; the *Firmicutes/Bacteroidetes* ratio in the gut decreased (2.12/1.95), while it increased in the lungs (5.46/10.17) (Fig. 1B). Through Linear Discriminant Analysis Effect Size (LEfSe) analysis, we further elucidated the differences in microbial communities between groups. In the gut, microbial communities characterized by high levels of *Bifidobacterium breve*, *Lactobacillus gasser*,

and *Selenomonadaceae* underwent significant changes. However, in the PAH-CHD group, levels of *Dialister*, *Streptococcus*, and *Roseburia inulinivorans* were significantly decreased (Fig. 1C). Compared to the control group, the PAH-CHD group had higher levels of *Enterococcus*, *Streptococcus peroris*, *Rothia mucilaginosa*, and *Roseburia inulinivorans* in lung microbiota, while levels of *Thermoactinomyces*, *Leptotrichia*, *Neisseria lactamica*, and *Streptococcus parasanguinis* were lower (Fig. 1D).

To further explore the microbial taxa that play a key role in inter-group differences, we constructed a random forest model at the species level and selected the top 10 species in inter-group comparison based on MeanDecreaseAccuracy and MeanDecreaseGin (Fig. 2A). In the gut, key microbial taxa include *Dialister invisus*, *Roseburia inulinivorans*, *Bacteroides stercoris*, *Prevotella timonensis*, and *Lactobacillus ruminis*, while in the lungs, the main differential microbial taxa are *Roseburia inulinivorans*, *Streptococcus peroris*, *Bacteroides fragilis*, and *Rothia mucilaginosa*. To assess the value of key gut and lung microbial taxa as biomarkers, we performed cross-validation in both training and validation sets and plotted receiver operating characteristic (ROC) curves. The results showed that the area under the ROC curve (AUC) was greater than 0.7 for both (Fig. S2). Additionally, we constructed genus-level co-abundance network maps of gut and lung microbiota in the two groups separately (Fig. 2B). The complexity of the gut microbial co-abundance network in the PAH-CHD group was found to be lower than that of the control. In the PAH-CHD group, *Ralstonia*, *Neisseria*, and *Prevotella* exhibited higher centrality, indicating a more simplified interaction between microbial communities compared to the control group. Interestingly, in BALF samples of the PAH-CHD group, the network complexity was higher than that in the control group. In the PAH-CHD group, *Bifidobacterium*, *Bacteroides*, and *Methyloversatilis* were identified as major hubs or key taxa.

To further explore the relationships between various taxa such as bacteria, fungi, and viruses, we conducted metagenomic analysis on a subset of samples. First, we investigated the composition and diversity of species in the gut and lungs between the two groups and observed increased α-diversity in the PAH-CHD group (Fig. S3A). Furthermore, PCoA of bacterial composition also showed partial separation between the two groups (Fig. S3B). Subsequently, we identified 18 and 23 differentially abundant species at the genus level in the gut and lungs, respectively, between the PAH-CHD and control groups. In the gut microbiota, compared to the control group, PAH-CHD patients exhibited enrichment of *Bifidobacterium*, *Hathewaya*, and *Astraeus*, while *Tyzzerella* and *Arabiibacter* were depleted. Meanwhile, in the lungs, *Campylobacter*, *Streptococcus*, *Clostridioides*, *Sanguibacteroides*, *Streptosporangium*, and *Microbacterium* were enriched, while *Acinetobacter*, *Mycobacterium*, *Burkholderia*, and *Klebsiella* were depleted (Fig. S3C). To facilitate the interpretation of microbial functions associated with PAH-CHD, we aggregated KEGG orthology (KO) level associations into Kyoto Encyclopedia of Genes and Genomes (KEGG) modules and performed Wilcoxon rank-sum tests (Fig. S3D). For the gut microbiota, enriched modules in the PAH-CHD group mainly involved energy metabolism, such as M00283, M00004, M00008, M00128, M00643, and M00835, while missing modules included M00079, M00532, M00127, and M00014. In the lung microbiota, PAH-CHD patients exhibited enrichment of modules related to energy metabolism, such as M00052, M00122, M00168, and M00122, while multiple modules associated with bacterial transport systems showed decreased expression. Subsequently, through KEGG functional enrichment analysis, we identified the active biological processes expressed in the PAH-CHD group. Interestingly, there was no difference in the functions identified by the gut microbiota between the groups, while the lung microbiota exhibited active expression in certain reaction pathways of amino acid biosynthesis, and the differentially annotated enzymes were widely distributed across various metabolic pathways, reflecting the complex energy metabolism and immune regulation in the lungs of PAH-CHD patients.

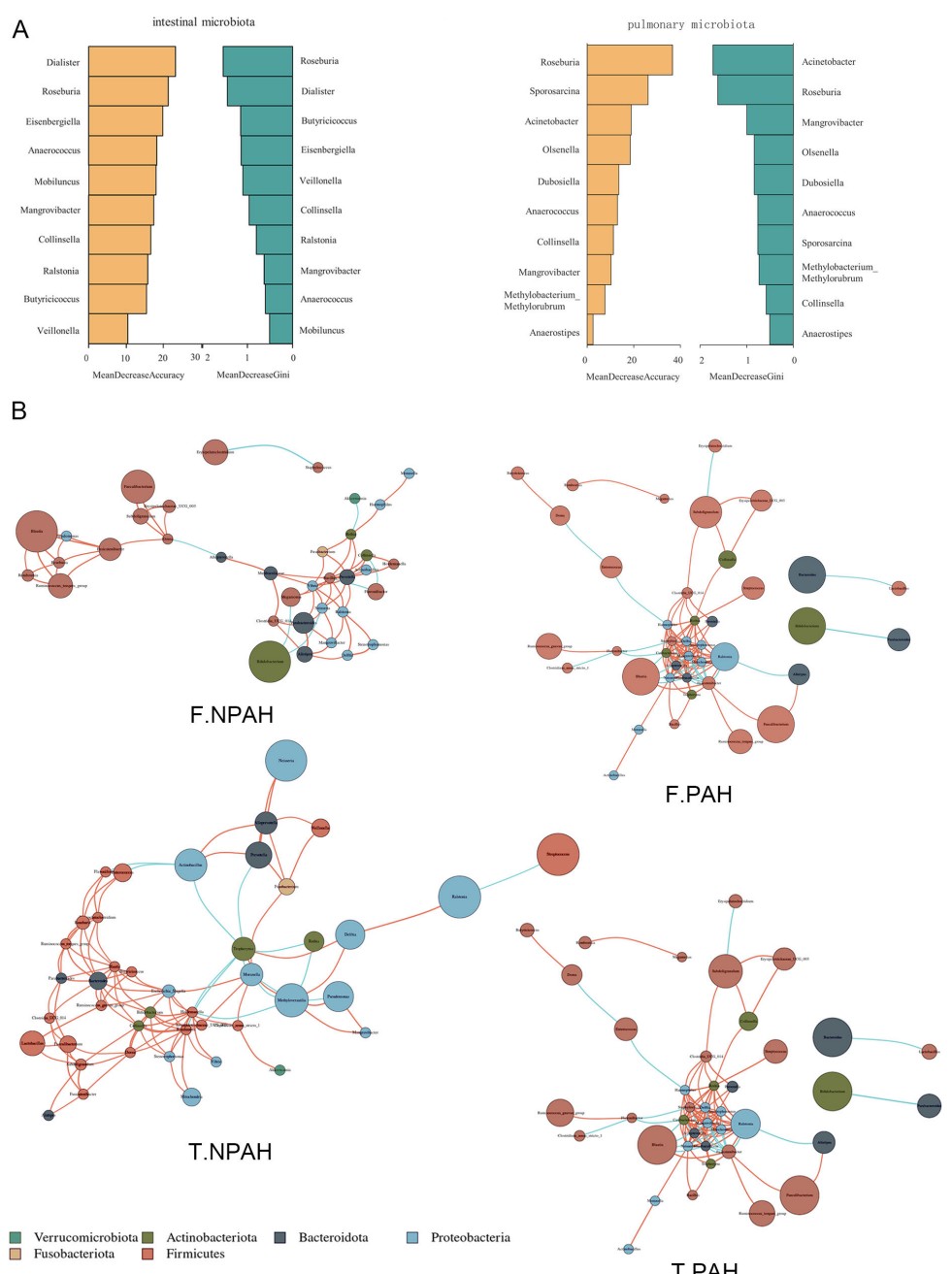

**FIG 2** Modeling to predict the community structure and dominant species of the microbial environment. (A) Random forest analysis identifies the most discriminative microbial genera in the gut and lungs between PAH-CHD and healthy control group. (B) Network diagram illustrating the correlations among the most abundant species (top 50) in the gut and lungs between PAH-CHD and healthy control group, with orange lines indicating positive correlations and blue lines indicating negative correlations (Spearman correlation test, $P < 0.05$, $R > 0.6$). PAH: PAH-CHD groups; NPAH: healthy controls; F-: derived from fecal samples; T-: derived from BALF samples.

## Metabolic profiling changes in PAH-CHD patients

We conducted a semi-targeted metabolomics analysis using liquid chromatography-mass spectrometry (LC-MS) on BALF, feces, and blood samples to characterize the metabolic differences between the two groups. The annotation results of metabolites in the three types of samples mainly include lipids, amino acids and their derivatives, and organic acids and their derivatives, accounting for approximately 65% of all

metabolites. PCoA analysis of metabolite distribution indicates partial differences in the overall metabolic characteristics between the two groups (Fig. 3A). In fecal samples, we identified a total of 789 metabolites. Through orthogonal partial least squares discriminant analysis, we identified 82 metabolites that differed between the PAH-CHD and control groups. Compared to the control group, the PAH-CHD group exhibited an upregulation of 30 metabolites, primarily bile acids, carnitines, amino acids, and their derivatives, while showing downregulation of 52 metabolites, primarily organic acids represented by citric acid and isocitric acid, and fatty acyls (Fig. 3B). In BALF samples, we identified a total of 502 metabolites, with 37 metabolites showing intergroup differences. Compared to the control group, the PAH-CHD group exhibited an upregulation of 25 metabolites, primarily bile acids, while showing downregulation of 12 metabolites, primarily organic acids represented by citric acid and isocitric acid, and sugars and their derivatives (Fig. 3B). In blood samples, we identified 35 differentially expressed metabolites, with 11 metabolites upregulated and 24 downregulated in the PAH-CHD group. Upregulated metabolites mainly involve metabolic pathways such as D-glucuronic acid, D-glucuronic acid, and aminomethylphosphonic acid, while downregulated metabolites mainly include tryptophan metabolism products such as 5-hydroxytryptamine, 2-indoleketone, uric acid, and indole-3-propionic acid (Fig. 3B). Among these, metabolites enriched in both the intestine and lungs mainly include bile acids such as Taurochenodeoxycholic acid and Taurodeoxycholic acid, while metabolites commonly downregulated in both locations mainly include fatty acyls, organic acids and their derivatives, and sugars and their derivatives. However, metabolites upregulated in the blood did not show consistent results in the intestinal and pulmonary metabolites (Fig. 3C).

## Microbiota and metabolomics correlation analysis in PAH-CHD patients

Metabolite origin analysis of differential metabolites and their involved metabolic pathways in each sample was conducted utilizing the MetOrigin platform (Fig. 4). Subsequently, we performed a Spearman correlation analysis of the microbiota and metabolites at the genus and species levels (Fig. S4 and S5). Among them, microbial communities such as *Bifidobacterium animalis*, *Bifidobacterium breve*, *Clostridium butyricum*, and *Lactobacillus gasseri* showed a positive correlation with primary bile acids and carnitines, reflecting the presence of active primary bile acid biosynthesis and fatty acid oxidation (FAO) processes (Fig. S5A). Furthermore, we generated BIO-Sankey, STA-Sankey, and network summary diagrams, which illustrate the correlations between microbial communities and metabolites at different taxonomic levels, and identified key metabolic pathways associated with pulmonary arterial hypertension (Fig. S6). The microorganisms were found to be actively involved in processes such as the tricarboxylic acid cycle, purine biosynthesis, and glycolysis/gluconeogenesis. Conversely, a negative correlation was observed with lysine degradation, arginine biosynthesis, linoleic acid metabolism, and the biosynthesis of unsaturated fatty acids and other related processes.

Our research took a unique approach by applying the same analytical process to the lung microbiota and metabolites. Through functional enrichment analysis of metabolites, we found that the identified 22 pathways overlapped with the results of fecal metabolite functional enrichment, mainly involving the citric acid cycle with isocitrate, citrate, and succinate as major participants, acetic acid and dicarboxylic acid metabolism, as well as primary bile acid biosynthesis and purine biosynthesis with taurochenodeoxycholic acid and glycochenodeoxycholic acid as participants (Fig. 4B). Subsequently, correlation analysis with differential lung microbiota revealed increased abundance of intestinal resident bacteria such as *Roseburia intestinalis* and *Roseburia inulinivorans*, actively involved in metabolic reactions, which positively correlated with differential metabolites such as Taurochenodeoxycholic acid, Taurolithocholic acid, and Glycodeoxycholic acid (Fig. S5B). The results of the intestinal and lung analyses were combined to identify similarities and differences in the differential metabolic pathways. The pathways were similarly characterized in both, with up-regulation of bile acid

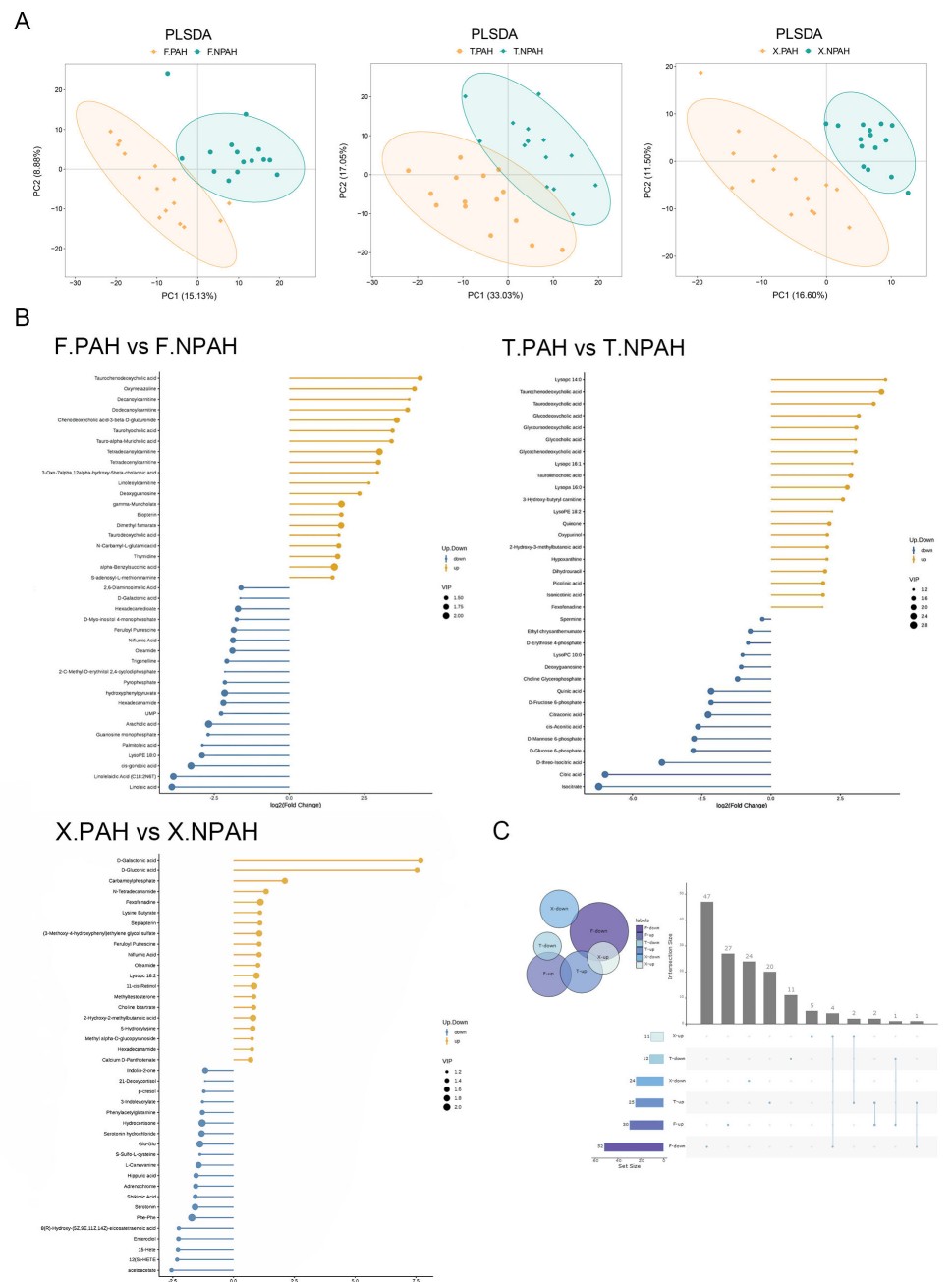

**FIG 3** Metabolomic profiles of the gut, lungs, and blood in children with PAH-CHD and healthy controls. (A) Partial least squares discriminant analysis (PLS-DA) reveals significant differences in metabolites between PAH-CHD and healthy control groups [variable importance in projection (VIP) > 1.0, FC > 2 or FC < 0.5, and $P$ < 0.05]. (B) Bar chart of the top 20 differentially upregulated and downregulated metabolites between PAH-CHD and healthy control groups, with blue bars indicating downregulation and yellow bars indicating upregulation. (C) Upset plot illustrating the intersection of upregulated and downregulated metabolites in the gut, lungs, and blood. PAH: PAH-CHD groups; NPAH: healthy controls; F-: derived from fecal samples; T-: derived from BALF samples; X-: derived from blood samples.

metabolism. However, the metabolic responses related to sugars, fats, and amino acids were differently active and closely associated with microorganisms (Fig. S7). This may reflect the host's active metabolic regulatory state in response to pathological changes.

Finally, the joint analysis of blood metabolites with intestinal and lung microbiota revealed that the highly expressed metabolite carbamoylphosphate in the PAH-CHD

group is involved in various amino acid metabolic pathways such as nitrogen metabolism, arginine synthesis, alanine, aspartate, and glutamate metabolism. It was shown to have a positive correlation with the abundance of upregulated microbial communities within the group, further validating the higher host metabolic rate in PAH-CHD patients (Fig. S8).

## Correlation study of metabolites with cardiac function and growth development in PAH-CHD patients

Spearman correlation analysis was conducted between differential metabolites in the intestine and lungs and clinical observation values to identify metabolites with greater clinical diagnostic and therapeutic significance. In the PAH-CHD group, intestinal metabolites such as 3-methylxanthine ($R = 0.63$), deoxycholic acid ($R = 0.55$), and arachidic acid ($R = 0.53$) showed positive correlations with S', while carnitines such as dodecanoylcarnitine ($R = -0.74$), tetradecenylcarnitine ($R = -0.61$), and linoleoylcarnitine ($R = -0.54$) showed negative correlations with S', with most substances corresponding positively to pulmonary artery systolic pressure (PASP) and ventricular septal defect (VSD). Growth and development were positively correlated with 1-methylnicotinamide ($R = 0.79$), allopurinol ($R = 0.75$), inosine ($R = 0.66$), and 1,4-naphthoquinone ($R = 0.65$). Conversely, deoxyguanosine ($R = -0.57$), dimethyl fumarate ($R = -0.56$), thymidine ($R = -0.56$), dimethylmalonic acid ($R = -0.53$), phenylpyruvic acid ($R = -0.53$), ethyl myristate ($R = -0.53$), and alpha-ketocaproic acid ($R = -0.52$) were negatively correlated, with these substances widely involved in metabolic pathways (Fig. 5A; Table S2). In lung metabolites, PASP showed positive correlations with taurochenodeoxycholic acid ($R = 0.65$), taurolithocholic acid ($R = 0.59$), glycoursodeoxycholic acid ($R = 0.55$), Lysopc 10:0 ($R = 0.65$), and Lysopc 14:0 ($R = 0.63$), among other bile acids and glycerophospholipids. Most metabolites also showed negative correlations with S', consistently representing right heart function. Moreover, the corresponding metabolites showed positive correlations with CI and negative correlations with growth and development (Fig. 5B; Table S3). Several amino acid metabolites in the blood, such as indoxylsulfuric acid, hippuric acid, and carbamoylphosphate, were negatively correlated with markedly elevated pulmonary arterial pressure and diminished right ventricular contractility (Fig. 5C; Table S4). These results suggest that different metabolites may be associated with right heart function and also affect the growth and development of patients. Combined with differential enrichment analysis of metabolites between groups, the regulation of metabolite levels fully reflects the body's ability to resist pathological changes. However, in this study, there were no specific metabolites correlated with ejection fraction (EF).

## DISCUSSION

To investigate the relationship between gut and lung microbiota and disease progression in children with PAH-CHD, we selected bronchoalveolar lavage fluid and fecal samples to explore the characteristics and associations of gut and lung microbes and metabolites between the patient group and healthy controls, considering the "gut-lung axis" (10–12). Further analysis showed that alterations in metabolites correlate with clinical indicators of pulmonary arterial hypertension. To our knowledge, this study represents the inaugural investigation into microbial-host interactions from a gut-lung axis perspective in children with PAH-CHD. Additionally, we enhanced our knowledge of microbial communities using metagenomic sequencing (10). Simultaneously, we advise readers that metagenomic sequencing technologies have both advantages and limitations; therefore, results should be interpreted carefully (13, 14).

The gut-lung axis is one of the pathways for communication between the gut and the lungs. On the one hand, microorganisms can colonize the lungs from the gut through microbial displacement, utilizing gut-draining lymphatic vessels, the portal venous system, systemic circulation, and microaspiration of oropharyngeal or oral microbiota into the respiratory and gastrointestinal tracts (15–22). Conversely, soluble microbial components, metabolites, and immune cells may enter the bloodstream and

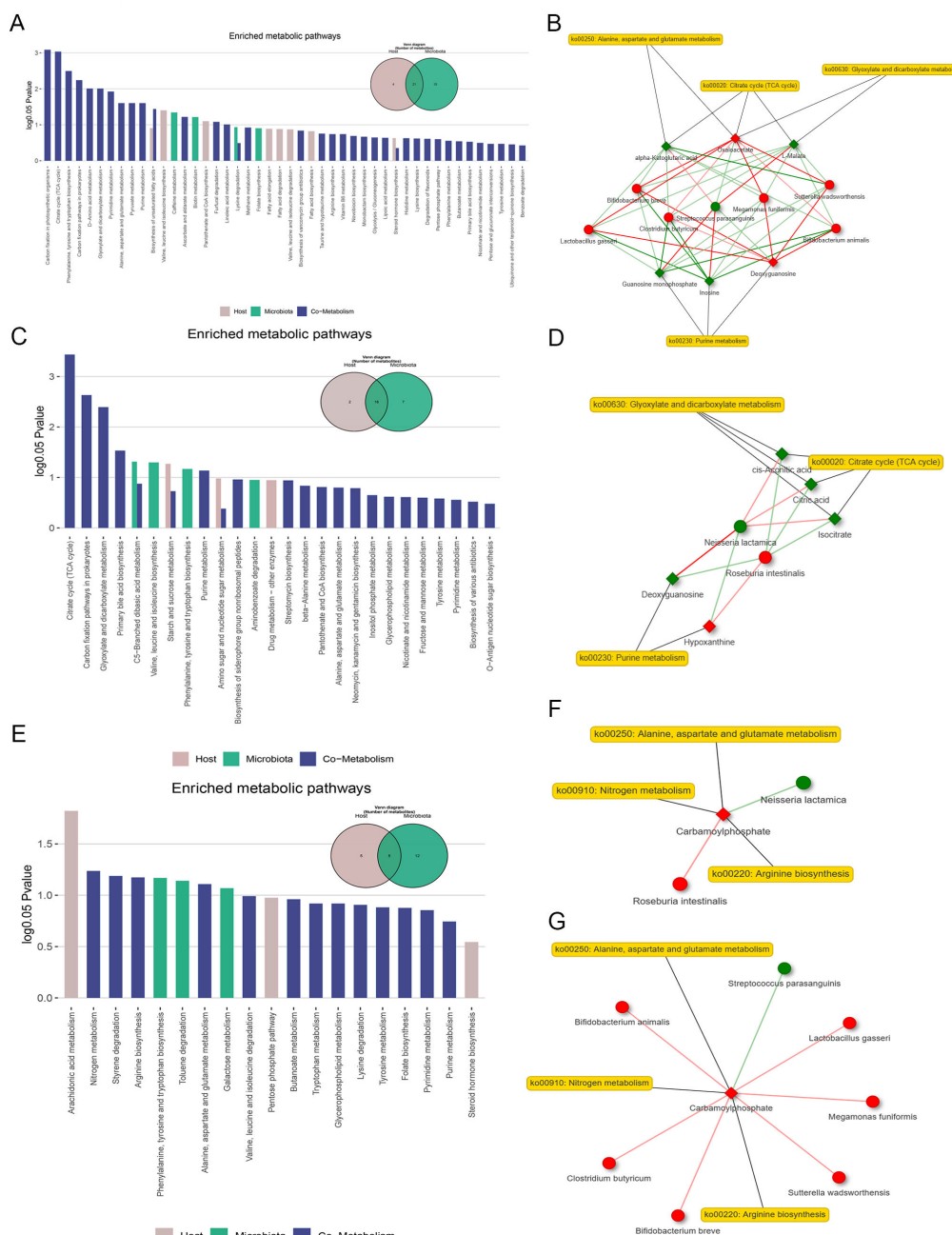

**FIG 4** Characteristic associations between microbiota and metabolites. (A) Statistical significance comparison of differentially sourced metabolites in the gut and their corresponding KEGG pathways (hypergeometric test, log0.05 *P* > 1). (B) Network summary analysis diagram of gut microbiota, metabolites, and metabolic pathways. (C) Statistical significance comparison of differentially sourced metabolites in the lung and their corresponding KEGG pathways (hypergeometric test, log0.05 *P* > 1). (D) Network summary analysis diagram of lung microbiota, metabolites, and metabolic pathways. (E) Statistical significance comparison of differentially sourced blood metabolites and their corresponding KEGG pathways (hypergeometric test, log0.05 *P* > 1). (F) Network summary analysis diagram of lung microbiota, blood metabolites, and metabolic pathways. (G) Network summary analysis diagram of gut microbiota, blood metabolites, and metabolic pathways. In the network summary analysis diagrams, diamonds and circles represent associated metabolites and microorganisms. Red (or green) nodes indicate populations with significantly higher (or lower) abundance. Red (or green) connections represent positive (or negative) correlations between microorganisms and metabolites. The related analyses were performed based on the MetOrigin platform.

lymphatic circulation via the gut-lung axis, potentially disrupting the balance of lung microbiota and influencing physiological and pathological processes in the lungs (23). Our analysis of lung microbiota revealed significant enrichment of gut symbionts such as *Roseburia* and *Enterococcus* in the PAH-CHD group. The differential microbial co-occurrence network indicated a positive correlation between these bacteria. In children with PAH-CHD, the Qp/Qs ratio exceeds 1.5, leading to prolonged pathological stimuli from high pressure and high volume in the pulmonary circulation, which modifies the survival environment of lung microbial communities and fosters microbial ectopic colonization. Consequently, gastrointestinal microbes are more prone to colonize the lungs through oropharyngeal microaspiration (24). Among them, *Roseburia*, particularly *Roseburia intestinalis*, plays an important role in preventing infections and alleviating and reversing pathological processes (25). A study in Sweden found that *Roseburia* levels were lower in patients with atherosclerosis compared to healthy subjects. Subsequent research indicated that in mouse models colonized with *Roseburia intestinalis*, the presence of *Roseburia intestinalis* was negatively correlated with the development of atherosclerotic plaques. It also participated in reprogramming from glycolysis to fatty acid metabolism, thereby reducing systemic inflammatory responses (26, 27). The growth of *Roseburia intestinalis* is influenced by carbon sources, symbionts, hypoxia, and pH levels. It can utilize various carbohydrates as energy sources to support its competitive advantage in the niche (25). Our study found that *Roseburia intestinalis* is associated with various energy metabolic activities, likely due to its production of butyrate by breaking down

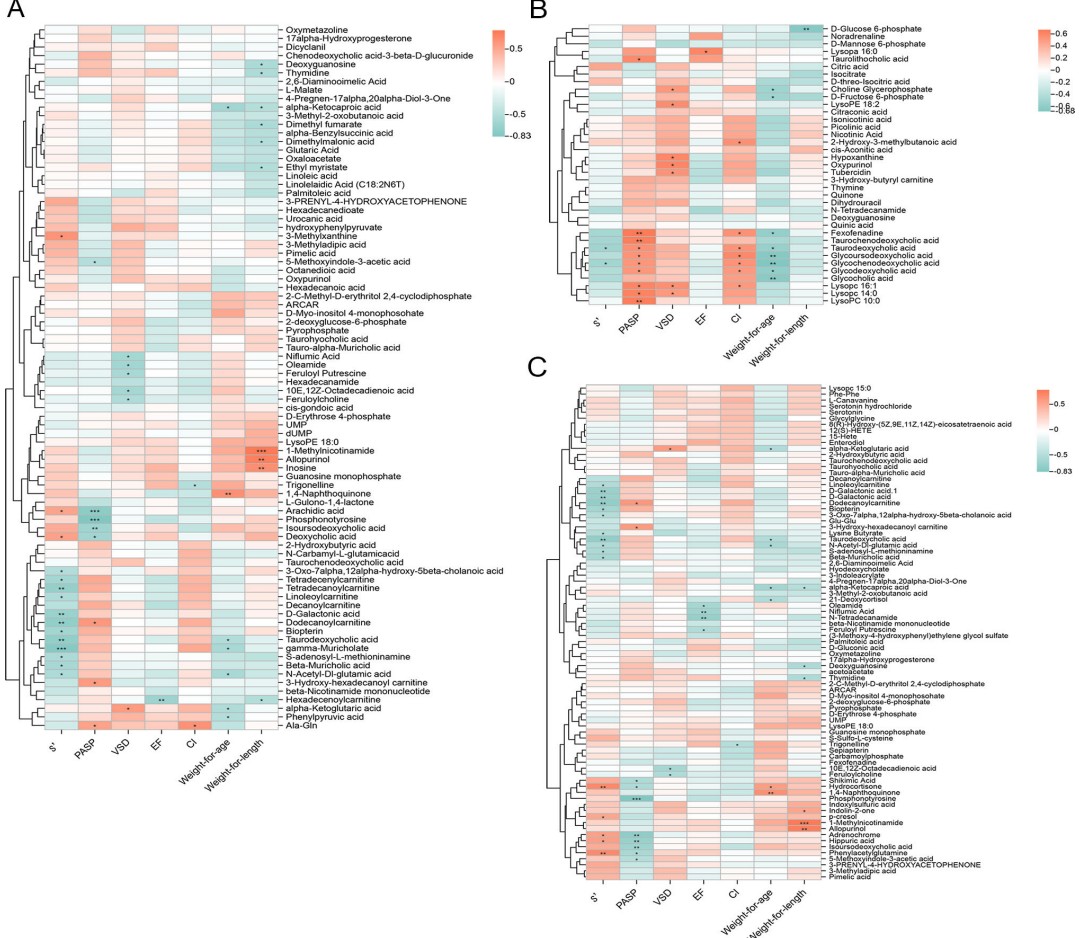

**FIG 5** Correlation between metabolites from gut (A), lung (B), and blood (C) and clinical evaluation indicators in PAH-CHD patients. Correlations were analyzed using Spearman's correlation analysis. Blue indicates positive correlation, and orange indicates negative correlation. Wilcoxon rank-sum test was used, with Benjamini-Hochberg false discovery rate multiple testing correction (FDR; *P* < 0.05). **P* < 0.05, ***P* < 0.01, and ****P* < 0.001.

different carbohydrates, which in turn exerts broad anti-inflammatory and metabolic regulatory effects (28–31). *Enterococcus* is a commensal bacterium of the human gastrointestinal tract that can also be pathogenic. It can colonize and survive in the host through virulence components such as gelatinase and phosphotransferase systems, and it can produce pathogenic effects such as inhibition of complement-mediated responses and biofilm formation (32–34). However, the role of *Enterococcus* in PAH-CHD is not yet clear, but its long-term colonization in the respiratory tract may increase the risk of opportunistic infections. Additionally, we found that oropharyngeal symbionts, such as *Streptococcus peroris*, *Streptococcus parasanguinis*, and *Neisseria lactamica*, differed in distribution between the PAH-CHD group and controls. These health-associated species were not absent in PAH-CHD children but had a reduced proportion in the overall community. These changes in microbial diversity might result from pathogenic factors disrupting the host's immune defenses, leading to the re-establishment of a non-pathogenic commensal community through bacterial migration (35). Nevertheless, some studies have indicated that when disease progression results in significant alterations to the local microenvironment, it can result in the loss of certain species, thereby precipitating an "ecological catastrophe" between the symbiotic organisms and the host (36). Children with left-to-right shunt PAH-CHD are prone to recurrent colds or pneumonia due to sustained increases in pulmonary circulation volume. This alters the pulmonary microenvironment and continuously impacts the stability of the lung microbiome.

Metabolites are critical components in the interactions between microbiota and the host. Trace analysis of lower respiratory tract metabolites revealed that bile acid metabolism pathways are active in children with PAH-CHD and are positively correlated with *Roseburia intestinalis*, *Roseburia inulinivorans*, *Streptococcus peroris*, and *Rothia mucilaginosa*. Microbiota are involved in the biotransformation and modification of bile acids, and the composition of bile acids has been shown to regulate the structure of the microbiota, which may also be one of the factors contributing to the reconstitution of lung microbiota (37). It has been demonstrated that obeticholic acid reduces pulmonary IL-6 mRNA expression and attenuates right ventricular hypertrophy, pulmonary vascular remodeling, and pulmonary hypertension in a rat model of PAH with wild lily alkaloids (38). The expression of bile acid receptors in pulmonary artery endothelial cells can also influence the activity of endothelin-1 and endothelial nitric oxide synthase (39, 40). This indicates that the increased production of microbial-derived bile acids and their vasodilatory activity might be beneficial for PAH. Additionally, studies have found that the concentrations of SCFAs and secondary bile acid cycling in PAH patients differ from those in healthy and familial control groups. Changes in the gut microbiome may be the underlying mechanism leading to these differences (41). In our study, we also found that bile acid metabolism pathways are actively expressed in the gut and positively correlate with gut-enriched *Bifidobacterium breve*, *Bifidobacterium animalis*, *Clostridium butyricum*, and *Lactobacillus gasseri*. The gut microbiota influences the development of pulmonary arterial hypertension by regulating bile acid concentrations.

*Bifidobacterium* and *Lactobacillus* are recognized as probiotics that effectively reduce inflammation and regulate immune responses and gut dysfunction by modulating the gut microbiota (42). Our study found that the gut microbiota of the PAH-CHD group is characterized by the enrichment of *Bifidobacterium*, *Lactobacillus*, *Clostridium*, and *Bacteroides*. The study found that *Bifidobacterium animalis* subsp. *lactis* XLTG11 enhances the intestinal barrier by upregulating tight junction proteins (ZO-1, Claudin-1, and Occludin) in the colon, while inhibiting the TLR4/NF-κB signaling pathway, increasing anti-inflammatory cytokines, and promoting the diversity and balance of gut microbiota, thereby strengthening host defenses (43). *Bifidobacterium breve* enhances the abundance of taxa related to bile acid regulation (such as *Bifidobacterium* and *Clostridium sensu stricto 1*), thereby promoting the uncoupling of bile acids in the intestine. In vitro cell experiments further confirmed that CA enhances the expression of tight junction proteins (Occludin and Claudin-1) and MUC2 (44). In the future, we can investigate further the effects of these strains on pulmonary tight junctions and inflammatory

responses, as well as how they may improve clinical outcomes in PAH-CHD patients. However, *Clostridium sensu stricto 1* is also an opportunistic pathogen associated with gut inflammation and can cause damage to intestinal epithelial cells (45). This suggests that the relationship between microbial community structure and host immune response is dynamic. Their ecological interactions determine whether local homeostasis is maintained or a transition to a disease state occurs (46).

In the analysis of differences in intestinal metabolites, we not only identified rich bile acid analogs but also observed an increase in carnosine analogs, which aligns with the research of Rhodes and Hongyang Pi et al. (47, 48). These substances, particularly carnosine, play a vital role in energy production in cardiac muscle cells (49). Carnitine homeostasis (acylcarnitine/free carnitine) is an important indicator for assessing mitochondrial function. Stephen M's research indicates that endothelial dysfunction in congenital heart disease models with increased pulmonary blood flow is associated with disruptions in carnitine homeostasis, mitochondrial dysfunction, decreased nitric oxide signaling, and enhanced reactive oxygen species. This finding has also been validated in patients with ventricular septal defects, suggesting that modulating carnitine homeostasis may become a new strategy for treating pulmonary vascular diseases (50–52). The high expression of bile acids and carnitine substances reflects the early self-protective mechanism initiated by the organism under the influence of pathogenic factors. The pathological process of PAH-CHD may remain in an adaptive state for a long time, and actively expressed protective factors are crucial for maintaining this state.

Notably, we observed an increase in the abundance of probiotics and elevated levels of bile acids and carnitine in the microbiota of children with PAH-CHD. These observations indicate that specific microbial communities and metabolic profiles may be linked to the development of pulmonary arterial hypertension. However, this does not suggest that they can be directly utilized as predictive markers for the disease. During the mild to moderate stages of pulmonary arterial hypertension, we hypothesize that the body may initiate protective mechanisms, with these probiotics and metabolites playing a crucial role in regulating interactions between local microbiota and the host, thereby helping to prevent or slow the progression of lesions (9, 53). We believe this may reflect a natural response by the body to maintain internal homeostasis and resist disease progression, a perspective also supported by the research of Rob Knight et al. (9). However, as the disease progresses, these protective factors may gradually decline. Coupled with the persistent pathological stimuli of high pressure and high volume in the pulmonary circulation, this ultimately results in irreversible damage in pulmonary arterial hypertension. Therefore, although changes in these microbial communities during the early stages of the disease may indicate a protective adjustment, they may also represent a point of dynamic change in the disease over time. Additionally, much of Yunnan Province is located at mid-altitude, with a diverse multi-ethnic population; the geographical environment and dietary habits may also influence the microbial community structure (9).

Another significant metabolic change we observed is the reprogramming of energy metabolism in the intestines and lungs of PAH-CHD patients, primarily manifested by the Warburg effect and enhanced fatty acid oxidation. The Warburg effect refers to the shift in cellular energy production from predominantly relying on mitochondrial oxidative phosphorylation to relying on aerobic glycolysis under conditions of sufficient oxygen (54). Aerobic glycolysis rapidly generates small amounts of ATP, providing intermediates for the pentose phosphate pathway for cell proliferation, and is influenced by tissue environment and stressors. For example, in conditions of hypoxia or angiogenesis promotion, endothelial cell glycolysis is enhanced, leading to increased lactate levels (55, 56). Therefore, the Warburg effect serves as a manifestation of mitochondria actively responding to the changes in hypoxic microenvironments induced by sustained pressure stimulation and inflammatory responses in pulmonary arterial hypertension. FAO is another critical source of energy, and its deficiency or excessive activation can disrupt energy metabolism, trigger inflammation and oxidative damage, disrupt immune

homeostasis, and lead to acute or chronic inflammatory diseases. Studies have found that mice lacking the malonyl-CoA decarboxylase (MCD) gene exhibit resistance to the development of pulmonary arterial hypertension during chronic hypoxia. MCD serves as a substrate for fatty acid synthesis enzymes and a regulator of FAO. Its deficiency hinders FAO, thereby inhibiting the development of PAH. Inhibiting FAO metabolism in endothelial cells may promote a metabolic shift toward protective glucose oxidation, enhancing endothelial cell tolerance to hypoxia (57). Therefore, we speculate that the tendency of glucose and lipid metabolism in the course of PAH may be a crucial therapeutic target. Additionally, our research findings suggest that the intestinal and pulmonary microbiota in the PAH-CHD group actively participate in arginine metabolism, which may be a compensatory enhancement. Nitric Oxide (NO) maintains vascular homeostasis and vasodilates vessels, and arginine is a substrate for NO synthesis. In pulmonary arterial hypertension models, inactivation of endothelial NO synthase in pulmonary arterial endothelial cells impedes the generation of NO from arginine within the cells (58). Concurrently, arginine metabolism and NO homeostasis in the intestine are associated with improving intestinal barrier dysfunction and alleviating inflammatory responses (59). Furthermore, dynamic amino acid metabolism responses appear to play a crucial role in the adaptation of the organism to changes in hemodynamics and the pulmonary pressure-flow relationship, affecting cardiopulmonary function (60).

We further investigated the correlation between metabolites and clinical prognostic indicators of pulmonary arterial hypertension, specifically focusing on their association with right heart function and the growth and development of pediatric patients. In children with PAH-CHD, substances such as bile acids and carnitines are highly enriched and closely associated with important indicators for assessing right heart function and prognosis, such as the peak systolic velocity S' measured at the junction of the tricuspid annulus (S'), PASP, and cardiac index (CI), demonstrating the body's active reparative response to pathological changes (61, 62). The decrease in tryptophan metabolites in the blood, such as indole, indoxyl sulfate, and 5-hydroxytryptamine, also reflects the body's protective effect on cardiopulmonary circulation (60). The conversion of arginine to urea is negatively correlated with mPAP measurements, consistent with our research findings (63). Furthermore, multiple studies have found associations between purine and pyrimidine-modified nucleosides and other metabolites with the phenotype and outcomes of PAH, with uric acid being a predictive factor for clinical deterioration in current research (47, 60, 64). Consistent changes in metabolites in the intestines and lungs reflect metabolic disturbances, which are correlated with growth and development. Among the included children, most exhibit poor growth and development. Congenital heart disease itself can lead to delayed growth and development in children, especially in those with pulmonary hypertension, where the condition is more severe (65). Therefore, in formulating treatment plans and implementing long-term follow-up processes, it is essential to fully consider the growth and development of children with congenital heart disease and to adopt appropriate exercise therapy measures to improve their overall health status (66).

This study identified commonalities between gut and lung microbial species and metabolites through bioinformatics analysis, but it has not adequately confirmed the role of the gut-lung axis in PAH-CHD. Therefore, future research should validate the key mechanisms underlying this hypothesis through fundamental experiments. Additionally, this study lacked an in-depth analysis of potential variables, including ethnicity, lifestyle background, and family factors, highlighting an area for further investigation.

Despite its limitations, our study highlights the following key points: (i) changes in the pulmonary microenvironment of PAH-CHD patients may facilitate adjustments in microbial dynamics; (ii) gut microbiota and their metabolites may trigger local pathological changes in PAH-CHD patients, which may relate to disease progression. These findings underscore the importance of maintaining microbial balance and implementing timely interventions in microbial communities for the treatment of PAH-CHD, which

may be crucial for restoring metabolic and immune homeostasis and could become a cornerstone of future therapeutic strategies.

## MATERIALS AND METHODS

### Patients and samples

The study subjects were children under 7 years old with left-to-right shunt congenital heart disease associated pulmonary arterial hypertension who were hospitalized at Fuwai Cardiovascular Hospital in Yunnan Province from June 2022 to January 2023 (15 cases). Exclusion criteria included (i) presence of other cardiovascular or pulmonary diseases, pulmonary complications, lung infections, metabolic diseases, and other systemic diseases; (ii) intake of probiotics, antibiotics, or immunosuppressants within 3 months prior to enrollment; (iii) participation in other clinical studies or inability to consent for nursing care or collection of BALF. Simultaneously, 15 matched healthy control subjects were recruited from Yunnan Children's Hospital. Sample and data collection obtained informed consent from all guardians of participants. Baseline demographic and clinical data were collected from electronic medical records.

### Analysis of 16S rRNA sequencing

Bacterial genomic DNA was extracted from BALF specimens using the Cetyltrimethylammonium Bromide (CTAB) method. Bacterial genomic DNA was extracted from stool specimens using the Magnetic Stool DNA Kit (TianGen, China, Catalog #: DP712). The purity and concentration of DNA were assessed using 1% agarose gel electrophoresis (AGE), and samples were diluted to 1 ng/µL with sterile water. Subsequently, PCR amplification of the hypervariable regions V3–V4 of the bacterial 16S rRNA gene was performed using primers 338F (5′-ACTCCTACGGAGGCAGCAG-3′) and 806R (5′-GGAC-TACHVGGGTWTCTAAT-3′). Library construction was carried out using the NEB Next Ultra II FS DNA PCR-free Library Prep Kit (New England Biolabs), followed by quantification using Qubit and Q-PCR, and paired-end 250 sequencing on the Illumina NovaSeq 6000 (Novogene, China). The raw tag data were filtered using fastp software (Version 0.23.1) and aligned and checked for chimeric sequences against species annotation databases (Silva database https://www.arb-silva.de/ for 16S/18S, Unite database https://unite.ut.ee/ for ITS), and chimeric sequences were removed to obtain the final valid data (67, 68). Denoising was performed using the DADA2 module in QIIME2 (Version QIIME2-202006) software to obtain the final amplicon sequence variants (ASVs) and feature table (69). Species annotation for each ASV was conducted using the classify-sklearn algorithm in QIIME2, employing a pre-trained Naive Bayes classifier (70, 71). Finally, normalization was performed based on the sample with the lowest data volume.

Alpha diversity was measured using the Chao1 and Shannon indices based on operational taxonomic units, with results visualized using Qiime2 (version qiime2-2020.6). To compare the abundance and diversity of bacteria among different samples, beta diversity was assessed by calculating the Bray-Curtis index, followed by visualization using principal coordinates analysis (PCoA), with results plotted using Qiime2 (version qiime2-2020.6). Significantly abundant bacteria were identified and classified using LEfSe, with visualization of LDA scores distribution and cladogram (LDA > 3.0, $P < 0.05$). A classification model was constructed using Random Forest algorithm (implemented in R using pROC and randomForest packages, Version 2.15.3) to identify important bacterial features associated with grouping. The model performance was evaluated using cross-validation and ROC curves, followed by selection of important species based on MeanDecreaseAccuracy and MeanDecreaseGini. Spearman correlation coefficient was calculated at genus level to generate a co-occurrence network diagram (parameters set as follows: removing connections with correlation coefficient <0.6; filtering out self-connections; removing connections with node abundance <0.005%).

## Metagenomic sequencing

DNA purity and concentration were accurately quantified using AGE and Qubit. Sequence libraries were generated using NEBNext Ultra DNA Library Prep Kit for Illumina (NEB, USA). The libraries were sequenced on the Illumina Hiseq X platform (insert size 350 bp, read length 150 bp) at the Novogene Bioinformatics Technology Co., Ltd. (Beijing, China). Low-quality reads were removed using Readfq (https://github.com/cjfields/readfq). Open Reading Frames (ORFs) were predicted from scaffolds ≥500 bp using MEGAHIT and MetaGeneMark (http://topaz.gatech.edu/GeneMark/), and predictions shorter than 100 nt were filtered out. Gene sequences were clustered into non-redundant gene catalogs at 95% identity and 90% coverage using CD-HIT software (http://www.bioinformatics.org/cd-hit/). The obtained gene set was aligned against the NCBI NR database (https://www.ncbi.nlm.nih.gov/) using DIAMOND software (https://github.com/bbuchfink/diamond/) to obtain species annotation information (parameters set as follows: blastp, evalue ≤1e-5).

Alpha diversity indices and PCoA were conducted based on different taxonomic levels using the R ade4 package. Subsequently, LEfSe analysis was employed to identify differentially abundant species at the genus level (LDA > 2, $P$ < 0.05). Unigenes were aligned against the KEGG database (http://www.kegg.jp/kegg/) using DIAMOND software (https://github.com/bbuchfink/diamond/) with parameters set as follows: blastp, -e 1e-5. Based on the alignment results, the relative abundances of different functional levels were calculated. LEfSe analysis was performed to identify inter-group functional differences at the KO level (LDA > 2, $P$ < 0.05). Finally, inter-group metabolic pathway comparison analysis was conducted.

## Targeted LC-MS metabolomics

Samples of BALF, feces, and blood for metabolomic analysis were thawed on ice after being removed from liquid nitrogen, and metabolites were extracted using methanol. Experimental samples were analyzed using multiple reaction monitoring based on the novoDB (Novogene database), with compounds quantified according to Q3 (daughter ion), and qualitatively analyzed based on Q1 (precursor ion), Q3 (daughter ion), retention time, declustering potential, and collision energy. Peak integration and calibration were performed using SCIEX OSV1.4 software, with peaks filtered based on settings such as a minimum peak height of 500, signal-to-noise ratio of 5, and smoothing points of 1. Identified metabolites were annotated using the KEGG database (https://www.genome.jp/kegg/pathway.html).

Metabolomic data were processed using the metaX software (72) for transformation, followed by partial least squares discriminant analysis to obtain the variable importance in projection (VIP) values for each metabolite. Statistical significance ($P$-values) of each metabolite between the two groups was calculated using a $t$-test, along with the calculation of fold change (FC) values between the two groups (parameter settings: VIP > 1.0, FC > 1.5 or FC < 0.667 with $P$-value < 0.05) (73–75). The correlation analysis between differential metabolites was conducted using the Pearson correlation coefficient in R language with the cor() function, and correlation plots were generated using the corrplot package in R language.

Differential metabolites were subjected to tracing and functional enrichment analysis using the MetOrigin platform (76). These metabolites were categorized into four groups through tracing analysis: host group (metabolites produced exclusively by the host), microbiota group (metabolites produced exclusively by the microbiota), co-metabolites group (metabolites produced by both the host and microbiota), and other group, including drug-related, food-related, environment-related, and unknown metabolites. Functional enrichment analysis was conducted based on their different sources. Furthermore, integrating differential microbial data, the correlation between differential metabolites and microbial taxa was initially revealed through Spearman analysis. Subsequently, Sankey network analysis was employed to visually illustrate the biological and statistical relationships between microbiota and metabolites. Finally,

network summarization analysis was performed to further elucidate the statistical and biological significance of the correlation between microbiota and metabolites in specific functional pathways.

## Clinical correlation analysis

Based on clinical data collected from electronic medical records, we investigated six core variables to assess cardiac function (tricuspid annular systolic velocity S', pulmonary artery systolic pressure, ventricular septal defect, ejection fraction, and cardiac index) in relation to growth and development (weight-for-age and weight-for-height Z-scores corresponding to percentiles). Variance Inflation Factor (VIF) for each variable was estimated using the vif function from the car package (https://www.example.com car/car.pdf) to address multicollinearity issues among the parameters. The correlation between each variable and differential metabolites in the gut, lungs, and blood was analyzed using the Spearman method. Chiplot (https://www.chiplot.online/) was utilized for calculation and visualization.

## Statistical analysis

Measurement data following a normal distribution were presented as mean ± SD, and differences between the two groups were evaluated using Student's $t$-test. Non-normally distributed data were analyzed using the non-parametric Mann-Whitney test. Spearman's correlation analysis is used to evaluate the relationship between two variables. Statistical analysis was conducted using GraphPad Prism software (version 8.0). A significance level of $P < 0.05$ was considered statistically significant. Additional statistical details regarding microbiome and metabolomics data can be found in the supplementary figures, legends, and methods.

## ACKNOWLEDGMENTS

Runwei Ma and Liming Cheng conceived and designed the experiments. Jiahui Xie, Xiaoyu Zhang, Yao Deng, and Haobo Ren performed experiments. Jiahui Xie and Xiaoyu Zhang analyzed the data. Jiahui Xie drafted the manuscript. Xiaoyu Zhang, Liming Cheng, Kai Liu, and Runwei Ma edited and revised the manuscript. All authors reviewed the content and approved the final version of the manuscript for publication.

The authors thank all the subjects who participated in this study.

This work was supported by "Xing Dian Elite Support Plan" of Yunnan Province—Foundation of Medical Specialist (No. YNWR-MY-2020–044) and Yunnan Provincial Clinical Research Center for Cardiovascular Diseases—New Technology Research and Development Project for Diagnosis and Treatment of Major Cardiovascular Diseases (No. 202102AA310002).

J.X.: conceptualization, data curation, formal analysis, methodology, visualization, writing—original draft, and writing—review and editing. X.Z.: conceptualization, data curation, formal analysis, methodology, project administration, and writing—review and editing. L.C.: conceptualization, data curation, formal analysis, methodology, and writing—review and editing. Y.D.: conceptualization, data curation, supervision, and writing—review and editing. H.R.: data curation, supervision, validation, and writing—review and editing. M.M., L.Z., C.M., and J.C.: supervision, validation, and writing—review and editing. K.L.: conceptualization, supervision, validation, and writing—review and editing. R.M.: conceptualization, funding acquisition, project administration, supervision, validation, and writing—review and editing.

## AUTHOR AFFILIATIONS

[1]Department of Cardiovascular Surgery, Fuwai Yunnan Hospital, Chinese Academy of Medical Sciences/Affiliated Cardiovascular Hospital of Kunming Medical University, Kunming, Yunnan Province, China

[2]Department of Cardiothoracic Surgery, The first hospital of Kunming, Kunming, Yunnan Province, China

[3]Department of Anesthesiology and Surgical Intensive Care Unit, Kunming Children's Hospital, Kunming, Yunnan Province, China

[4]Comprehensive Pediatrics, Kunming Children's Hospital, Kunming, Yunnan Province, China

## AUTHOR ORCIDs

Jiahui Xie http://orcid.org/0009-0002-0864-9165
Xiaoyu Zhang http://orcid.org/0000-0003-4603-2908
Kai Liu http://orcid.org/0000-0002-3374-3375
Runwei Ma http://orcid.org/0000-0002-2817-9351

## FUNDING

| Funder | Grant(s) | Author(s) |
| --- | --- | --- |
| Yunnan Provincial Health Commission | No. YNWR-MY-2020-044 | Runwei Ma |
| Fuwai Yunnan Hospital,Chinese Academy of Medical Sciences | No. 202102AA310002 | Runwei Ma |

## AUTHOR CONTRIBUTIONS

Jiahui Xie, Conceptualization, Data curation, Formal analysis, Methodology, Visualization, Writing – original draft | Xiaoyu Zhang, Conceptualization, Data curation, Formal analysis, Methodology, Project administration, Writing – review and editing | Liming Cheng, Conceptualization, Data curation, Formal analysis, Methodology, Writing – review and editing | Yao Deng, Conceptualization, Data curation, Supervision, Writing – review and editing | Haobo Ren, Data curation, Supervision, Validation, Writing – review and editing | Minghua Mu, Supervision, Validation, Writing – review and editing | Liang Zhao, Supervision, Validation, Writing – review and editing | Chunjie Mu, Supervision, Validation, Writing – review and editing | Jiaxiang Chen, Supervision, Validation, Writing – review and editing | Kai Liu, Conceptualization, Supervision, Validation, Writing – review and editing | Runwei Ma, Conceptualization, Funding acquisition, Project administration, Supervision, Validation, Writing – review and editing

## DATA AVAILABILITY

Raw sequencing data have been publicly deposited and are available at the NCBI Sequence Read Archive, with BioProject accession no. PRJNA1166969.

## ETHICS APPROVAL

This study was approved by The institutional review board of Fuwai Yunnan Hospital, Chinese Academy of Medical Sciences (Approval No: 2021–017-04) and adhered to the principles of the Declaration of Helsinki. Informed consent was obtained from all participants.

## ADDITIONAL FILES

The following material is available online.

### Supplemental Material

**Data Set S1 (Spectrum01808-24-s0001.xlsx).** Metabolite expression levels for each sample.
**Supplemental figures (Spectrum01808-24-s0002.pdf).** Figures S1 to S8.
**Supplemental tables (Spectrum01808-24-s0003.xlsx).** Tables S1 to S4.

Open Peer Review

**PEER REVIEW HISTORY (review-history.pdf).** An accounting of the reviewer comments and feedback.

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
