## [Reviewer comments · Microbiology Spectrum]

Microbiology Spectrum

Integrated Multi-Omics Analysis of the Microbial Profile Characteristics Associated with Pulmonary Arterial Hypertension in Congenital Heart Disease

Jiahui Xie, Xiaoyu Zhang, Liming Cheng, Yao Deng, Haobo Ren, Minghua Mu, Liang Zhao, Chunjie Mu, Jiaxiang Chen, Kai Liu, and Runwei Ma

Corresponding Author(s): Runwei Ma, Department of Cardiovascular surgery, Fuwai Yunnan Hospital, Chinese Academy of Medical Sciences/ Affiliated Cardiovascular Hospital of Kunming Medical University

Review Timeline:

Submission Date:	July 22, 2024
Editorial Decision:	September 7, 2024
Revision Received:	October 7, 2024
Accepted:	October 9, 2024

Editor: Benjamin Liu

Reviewer(s): Disclosure of reviewer identity is with reference to reviewer comments included in decision letter(s). The following individuals involved in review of your submission have agreed to reveal their identity: Hawraa Natiq Kabroot AL-Fatlawy (Reviewer #1)

Transaction Report:

DOI: <https://doi.org/10.1128/spectrum.01808-24>

Re: Spectrum01808-24 (Revealing through Multi-omics Analysis the Functional Characteristics of the "Gut-Lung Axis" Microbiome Associated with Congenital Heart Disease and Pulmonary Arterial Hypertension)

Dear Dr. Runwei Ma:

Thank you for the privilege of reviewing your work. Below you will find my comments, instructions from the Spectrum editorial office, and the reviewer comments.

Editor's comments:

1. Line 301-303 "We also employed metagenomic sequencing to enhance our understanding of the microbiome, building upon previous research findings". There are no references when mentioning previous research findings. Also, please add "Of note, metagenomic sequencing has its pros and cons that warrant careful interpretation of its results". More references should be cited, with the following ones as examples (citing is optional):

Epidemiological and clinical overview of the 2024 Oropouche virus disease outbreaks, an emerging/re-emerging neurotropic arboviral disease and global public health threat. *J Med Virol.* 2024 Sep;96(9):e29897. doi: 10.1002/jmv.29897. PMID: 39221481.

Laboratory diagnosis of CNS infections in children due to emerging and re-emerging neurotropic viruses. *Pediatr Res.* 2024 Jan;95(2):543-550. doi: 10.1038/s41390-023-02930-6. Epub 2023 Dec 2. PMID: 38042947.

2. Line 304-310: There are no references when mentioning previous research findings. More references should be cited, with the following ones as examples (citing is optional):

The Brief Case: Ventilator-Associated *Corynebacterium accolens* Pneumonia in a Patient with Respiratory Failure Due to COVID-19. *J Clin Microbiol.* 2021 Aug 18;59(9):e0013721. doi: 10.1128/JCM.00137-21. Epub 2021 Aug 18. PMID: 34406882; PMCID: PMC8372998.

The Brief Case: *Capnocytophaga sputigena* Bacteremia in a 94-Year-Old Male with Type 2 Diabetes Mellitus, Pancytopenia, and Bronchopneumonia. *J Clin Microbiol.* 2021 Jun 18;59(7):e0247220. doi: 10.1128/JCM.02472-20. Epub 2021 Jun 18. PMID: 34142857; PMCID: PMC8218739.

Please return the manuscript within 30 days; if you cannot complete the modification within this time period, please contact me. If you do not wish to modify the manuscript and prefer to submit it to another journal, notify me immediately so that the manuscript may be formally withdrawn from consideration by Spectrum.

Revision Guidelines

Data availability: ASM policy requires that data be available to the public upon online posting of the article, so please verify all links to sequence records, if present, and make sure that each number retrieves the full record of the data. If a new accession number is not linked or a link is broken, provide Spectrum production staff with the correct URL for the record. If the accession numbers for new data are not publicly accessible before the expected online posting of the article, publication may be delayed;

please contact production staff (Spectrum@asmusa.org) immediately with the expected release date.

Sincerely,
Benjamin Liu
Editor
Microbiology Spectrum

Reviewer #1 (Comments for the Author):

Reviewer Attachments for Manuscript Number ASM Spectrum01808-24
Revealing through Multi-omics Analysis the Functional Characteristics of the "Gut-Lung Axis" Microbiome Associated with Congenital Heart Disease and Pulmonary Arterial Hypertension

Dear Authors

There are several notes regarding your manuscript as follows.

Reorder your search as follows:

- 1- abstract
- 2- keywords
- 3- introduction
- 4- material and methods
- 5- results
- 6- discussion

.....
DISCUSSION

Please, add modern reference for discussion

293 In order to investigate the relationship between gut and lung flora and disease 294 progression in children with PAH-CHD, we chose alveolar lavage fluid and faeces 295 as biological samples. We explored the characteristics and associations of gut and 296 lung microbiota and metabolites from the perspective of the "gut-lung axis" in 297 PAH-CHD patients compared with healthy controls. Fur

Conclusion: add Conclusion with numbered

Bibliography/References

References

References are relevant to the study field and recent, it is recommended to use software like Mendeley or other to manage the bibliography list.

Reviewer #2 (Comments for the Author):

Specific comments:

1. The title of the manuscript suggests that authors have analyzed the functional characteristics of the gut-lung axis microbiome but functional characteristics described are mostly speculative. The authors have merely performed data mining to confirm the presence (or absence) of certain bacterial species in PAH-CHD patients as compared to normal subjects.
2. Kim et al (2020) suggested that the altered microbiome of patients with PAH could predict disease condition. However, in this manuscript the authors report that the gut and pulmonary microbiota of children with PAH-CHD is rich in 'beneficial symbionts'. This cannot be taken as a predictor of disease and seems contradictory.
3. In Figure 1B, the authors claim that the PCoA shows differences in the microbiota of the gut and lung. However, the Venn diagram shows more common species and less different ones. The different ones in the gut are all beneficial bacteria!

4. I did not see a strong evidence of 'gut-lung axis' involvement. KO analysis using KEGG modules to depict functional commonalities between the gut and lung microbiota is not sufficient to suggest the involvement of the gut-lung axis in PAH-CHD.

5. The authors speculate that microbial species may travel from the gut to the lung via blood (lines 313-315) but the metabolite analysis from blood did not corroborate with that from the gut and lung (Figure 2C).

6. Authors show that PAH-CHD samples are enriched in *Bifidobacterium animalis* which causes expression of aquaporins and tight junction proteins (lines 380-381). Did they check for gut permeability in gut epithelial cells of the patients? Are lung tight junctions affected?

Minor comments:

1. Line 26 states that 'Author order was determined on the basis of seniority'. Established criteria should be followed to determine the sequence of authors.

2. The title of the manuscript is misleading and needs correction in language.

3. In several places in the text, earlier works are referred to but the reference numbers are missing. Examples include line 83 ("In their 2020 study, Kim et al highlighted---") and line 398 ("Stephen M's research indicates -----")

4. I found most of the figures very cramped and difficult to read and understand. Many figure legends are not adequately described.

5. In some places, sentences are repeated. Example: lines 343-347 are same as lines 355-359 but the cited reference is different.

Reviewer Attachments for Manuscript Number ASM Spectrum01808-24

Revealing through Multi-omics Analysis the Functional Characteristics of the "Gut-Lung Axis" Microbiome Associated with Congenital Heart Disease and Pulmonary Arterial Hypertension

Dear Authors

There are several notes regarding your manuscript as follows.

Reorder your search as follows:

- 1- abstract
- 2- keywords
- 3- introduction
- 4- material and methods
- 5- results
- 6- discussion

.....

DISCUSSION

Please, add modern reference for discussion

293 In order to investigate the relationship between gut and lung flora and disease 294 progression in children with PAH-CHD, we chose alveolar lavage fluid and faeces 295 as biological samples. We explored the characteristics and associations of gut and 296 lung microbiota and metabolites from the perspective of the "gut-lung axis" in 297 PAH-CHD patients compared with healthy controls. Fur

Conclusion: add Conclusion with numbered

Bibliography/References

References

References are relevant to the study field and recent, it is recommended to use software like Mendeley or other to manage the bibliography list.

Mr. Editor, Thank you very much for choosing us to review and evaluate this paper, and I hope that we will have many contributions in the future. .With appreciation and respect to you

With best regards

**Assist. Prof. Dr. Hawraa Natiq Kabroot AL-Fatlawy
Faculty of Medicine , University of Kufa, Iraq
Ph.D (Microbiology / Molecular Genetics)
10 / 8 / 2024**

Review Comments

General comments:

In this manuscript, the authors have used a multi-omics approach to examine the microbiome in the gut and lung samples from patients diagnosed with PAH-CHD. Overall, I found the manuscript rather descriptive and lacking in novelty as a similar study was reported by Kim et al (Kim S et al. Altered Gut Microbiome Profile in Patients with Pulmonary Arterial Hypertension. Hypertension. 2020 April; 75(4): 1063–1071. doi:10.1161/HYPERTENSIONAHA.119.14294).

Specific comments:

1. The title of the manuscript suggests that authors have analyzed the functional characteristics of the gut-lung axis microbiome but functional characteristics described are mostly speculative. The authors have merely performed data mining to confirm the presence (or absence) of certain bacterial species in PAH-CHD patients as compared to normal subjects.
2. Kim et al (2020) suggested that the altered microbiome of patients with PAH could predict disease condition. However, in this manuscript the authors report that the gut and pulmonary microbiota of children with PAH-CHD is rich in ‘beneficial symbionts’. This cannot be taken as a predictor of disease and seems contradictory.
3. In Figure 1B, the authors claim that the PCoA shows differences in the microbiota of the gut and lung. However, the Venn diagram shows more common species and less different ones. The different ones in the gut are all beneficial bacteria!
4. I did not see a strong evidence of ‘gut-lung axis’ involvement. KO analysis using KEGG modules to depict functional commonalities between the gut and lung microbiota is not sufficient to suggest the involvement of the gut-lung axis in PAH-CHD.
5. The authors speculate that microbial species may travel from the gut to the lung via blood (lines 313-315) but the metabolite analysis from blood did not corroborate with that from the gut and lung (Figure 2C).
6. Authors show that PAH-CHD samples are enriched in *Bifidobacterium animalis* which causes expression of aquaporins and tight junction proteins (lines 380-381). Did they check for gut permeability in gut epithelial cells of the patients? Are lung tight junctions affected?

Minor comments:

1. Line 26 states that ‘Author order was determined on the basis of seniority’. Established criteria should be followed to determine the sequence of authors.
2. The title of the manuscript is misleading and needs correction in language.
3. In several places in the text, earlier works are referred to but the reference numbers are missing. Examples include line 83 (“In their 2020 study, Kim et al highlighted---”) and line 398 (“Stephen M’s research indicates -----”)
4. I found most of the figures very cramped and difficult to read and understand. Many figure legends are not adequately described.
5. In some places, sentences are repeated. Example: lines 343-347 are same as lines 355-359 but the cited reference is different.

Response to reviewers

Dear editor and reviewers.

Thank you for offering us an opportunity to improve the quality of our submitted manuscript (Spectrum01808-24). We appreciated very much the reviewers' constructive and insightful comments. In this revision, we have addressed all of these suggestions. We hope the revised manuscript now meets the publication standard of your journal. We highlighted all the revisions in yellow colour.

On the next pages, our point-to-point responses to the queries raised by the reviewers are listed.

Editor's comments

Comment 1: Line 301-303 "We also employed metagenomic sequencing to enhance our understanding of the microbiome, building upon previous research findings". There are no references when mentioning previous research findings. Also, please add "Of note, metagenomic sequencing has its pros and cons that warrant careful interpretation of its results". More references should be cited, with the following ones as examples (citing is optional):

Epidemiological and clinical overview of the 2024 Oropouche virus disease outbreaks, an emerging/re-emerging neurotropic arboviral disease and global

public health threat. *J Med Virol.* 2024 Sep;96(9):e29897. doi: 10.1002/jmv.29897. PMID: 39221481.

Laboratory diagnosis of CNS infections in children due to emerging and re-emerging neurotropic viruses. *Pediatr Res.* 2024 Jan;95(2):543-550. doi: 10.1038/s41390-023-02930-6. Epub 2023 Dec 2. PMID: 38042947.

Response: Thank you for your thorough review. We acknowledge that in the previously submitted version, we did not cite the relevant literature to support the statement, "We employed metagenomic sequencing to deepen our understanding of microbial communities," which was an oversight on our part. We apologize for this error. Based on your suggestion, we have added reference [10] to the revised manuscript. Your suggestion that "metagenomic sequencing has both advantages and limitations, and its results should be interpreted with caution" is helpful for guiding readers to objectively assess the research findings. We have integrated this perspective into the revised manuscript. Additionally, we appreciate the references you provided, and we have included them in the revised manuscript as [13] and [14].

The specific modifications are as follows: "Additionally, we enhanced our knowledge of microbial communities using metagenomic sequencing(10). Simultaneously, we advise readers that metagenomic sequencing technologies have both advantages and limitations; therefore, results should be interpreted carefully(13, 14)." (Page 16, lines 300-303.)

Comment 2: Line 304-310: There are no references when mentioning previous research findings. More references should be cited, with the following ones as examples (citing is optional):

The Brief Case: Ventilator-Associated *Corynebacterium accolens* Pneumonia in a Patient with Respiratory Failure Due to COVID-19. *J Clin Microbiol.* 2021 Aug 18;59(9):e0013721. doi: 10.1128/JCM.00137-21. Epub 2021 Aug 18. PMID: 34406882; PMCID: PMC8372998.

The Brief Case: *Capnocytophaga sputigena* Bacteremia in a 94-Year-Old Male with Type 2 Diabetes Mellitus, Pancytopenia, and Bronchopneumonia. *J Clin Microbiol.* 2021 Jun 18;59(7):e0247220. doi: 10.1128/JCM.02472-20. Epub 2021 Jun 18. PMID: 34142857; PMCID: PMC8218739.

Response: Thank you for your thorough review and the references you provided. We have revised the aforementioned sentences based on the reviewers' suggestions and added relevant references. Identifying the relationship between microorganisms and pulmonary diseases from clinical cases is a valuable suggestion with significant clinical implications. We also reviewed several research papers on the "gut-lung axis" and found evidence that directly discusses their connection. Following your suggestions, we have made the following specific modifications in the revised manuscript:

“On the one hand, microorganisms can colonize the lungs from the gut through microbial displacement, utilizing gut-draining lymphatic vessels, the portal venous system, systemic circulation, and microaspiration of oropharyngeal or oral microbiota into the respiratory and gastrointestinal tracts(15-22). Conversely, soluble microbial components, metabolites, and immune cells may enter the bloodstream and lymphatic circulation via the gut-lung axis, potentially disrupting the balance of lung microbiota and influencing physiological and pathological processes in the lungs(23).” (Page 16, lines 306-313.)

Reviewer #1

Comment 1: Reorder your search as follows:

1- abstract

2- keywords

3- introduction

4- material and methods

5- results

6- discussion

Response: Thank you for your valuable suggestions. We understand that your suggestions aim to enhance the article's fluency and readability. Before submitting the current manuscript, we referred to the journal's formatting guidelines for typesetting and formatting. In the typesetting of the original

manuscript, we considered the following points:

1. The abstract and keywords are placed at the beginning of the article, conforming to the standard structure of scientific papers, which allows readers to quickly grasp the key research points.
2. The introduction follows, providing the necessary research background and literature review, laying the foundation for understanding our motivation and objectives.
3. Placing the results section before the methods emphasizes the innovation and significance of the research, enabling readers to quickly assess whether the article aligns with their research interests and fostering interdisciplinary communication.
4. The discussion section follows the results, providing an in-depth analysis of the findings.
5. The materials and methods section is placed at the end, underscoring the importance of the results while allowing non-specialist readers to quickly access essential information. Additionally, according to the journal's guidelines, the materials and methods section is excluded from the word count, allowing us to describe the experimental procedures in detail for other researchers seeking to replicate the study.

However, based on your suggestion, we are also submitting another version—"Manuscript(Revised Draft-Reviewer #1)"—for further review and structural adjustment by you and the journal's editorial team.

Comment 2: DISCUSSION

Please, add modern reference for discussion

In order to investigate the relationship between gut and lung flora and disease progression in children with PAH-CHD, we chose alveolar lavage fluid and faeces as biological samples. We explored the characteristics and associations of gut and lung microbiota and metabolites from the perspective of the "gut-lung axis" in PAH-CHD patients compared with healthy controls.

Response: Thank you for your valuable suggestions. Following your guidance, we have added several references in the discussion section to deepen the exploration of PAH-CHD, gut microbiota, and lung microbiota, and their relationship with disease progression, cited as [10], [11], and [12]. The details are as follows: 1) Ma R, *Front Med (Lausanne)*, 2022; 9:940784. (doi:10.3389/fmed.2022.940784) is a previous study that analyzed changes in the lower respiratory tract microbiota and metabolites of PAH-CHD, CHD, and healthy subjects through 16S rRNA and metabolomics, revealing that disrupted pulmonary microbiota and metabolites in PAH-CHD could serve as biomarkers. This study builds upon and expands our earlier research; 2) Wang T, *Hypertension*, 2023; 80(1): 214-226. (doi:10.1161/HYPERTENSIONAHA.122.19182) highlights the association between upper respiratory tract microbiota diversity and the severity of PAH-CHD, providing new insights into the relationship between microbiota

and PAH-CHD; 3) Yang Y, *Imeta*, 2024; 3(2): e159. (doi:10.1002/imt2.159) reviews the role of gut microbiota and their metabolites in pulmonary hypertension (PH), emphasizing the importance of studying the gut-lung axis. These references help readers understand the innovations in our research and the considerations regarding the selection of biological samples.

Comment 3 : Conclusion: add Conclusion with numbered

Response : Thank you for your valuable suggestions. Following your guidance, we have discussed the limitations and conclusions separately in the revised manuscript to enhance the structure and clarity of the article. Additionally, we have added numbering in the conclusions section to help readers better follow and understand the content. The specific modifications are as follows:

“Despite its limitations, our study highlights the following key points: 1) Changes in the pulmonary microenvironment of PAH-CHD patients may facilitate adjustments in microbial dynamics; 2) Gut microbiota and their metabolites may trigger local pathological changes in PAH-CHD patients, which may relate to disease progression. These findings underscore the importance of maintaining microbial balance and implementing timely interventions in microbial communities for the treatment of PAH-CHD, which may be crucial for restoring metabolic and immune homeostasis and could become a cornerstone of future therapeutic strategies.” (Page 26, lines 503-511.)

Comment 4 : Bibliography/References

References

References are relevant to the study field and recent, it is recommended to use software like Mendeley or other to manage the bibliography list.

Response: Thank you for your valuable suggestions. To facilitate the use of various citation and text editing software, we are currently using EndNote for reference management, ensuring the accuracy of the references. We have also explored Mendeley, as you recommended, and found it to excel in group collaboration. We have suggested to our project leader and team members considering Mendeley for reference management in the future, as it will facilitate teamwork and communication, and keep references updated during the research period.

Reviewer #2

Comment 1: The title of the manuscript suggests that authors have analyzed the functional characteristics of the gut-lung axis microbiome but functional characteristics described are mostly speculative. The authors have merely performed data mining to confirm the presence (or absence) of certain bacterial species in PAH-CHD patients as compared to normal subjects.

Response : Thank you for your valuable feedback and review. You correctly pointed out that our paper's title implies an in-depth analysis of the functional

characteristics of the gut-lung axis microbiota, while the actual content leans more toward data-driven speculation. This study primarily employed data mining methods to identify differences in specific bacterial taxa between PAH-CHD patients and healthy controls. We acknowledge that due to limitations in the study design, we were unable to perform experimental validation; thus, our conclusions regarding the functional characteristics of the gut-lung axis are speculative and based on literature review and data analysis. Our goal is to use data mining to generate hypotheses and research directions for future studies. We plan to further investigate these speculative conclusions through experimental methods in future research, validating whether these differences have biological significance and functional relevance. To more accurately reflect the content and methods of our study, we have adjusted the paper's title to better convey the nature of our research methods and findings. The revised title is: “Integrated Multi-Omics Analysis of the Microbial Profile Characteristics Associated with Pulmonary Arterial Hypertension in Congenital Heart Disease”.

Comment 2: Kim et al (2020) suggested that the altered microbiome of patients with PAH could predict disease condition. However, in this manuscript the authors report that the gut and pulmonary microbiota of children with PAH-CHD is rich in 'beneficial symbionts'. This cannot be taken as a predictor of disease and seems contradictory.

Response: Thank you for highlighting the potential differences between our findings and those of Kim et al. (2020), which indeed warrant further discussion.

Kim S, Hypertension, 2020; 75(4): 1063-1071. (doi:10.1161/HYPERTENSIONAHA.119.14294) revealed differences in the gut microbiome between adults with type I PAH and healthy controls through metagenomic analysis, finding specific gut microbiota closely associated with carnitine metabolism and the biosynthesis of arginine, ornithine, and proline. They achieved an accuracy of 83% in predicting PAH using a random forest model. Our study focuses on the gut and lung microbiota of children with PAH-CHD. Through species difference analysis and a random forest model, we identified a range of microbiota potentially associated with PAH-CHD. The observed increase in “beneficial symbionts” may relate to the physiological characteristics and disease stages of children.

The key microbiota we identified, such as *Dialister invisus* and *Roseburia inulinivorans*, differ from Kim et al.'s findings, which may reflect the diversity of microbiota composition across different disease stages and patient populations. Our findings provide a new perspective on understanding the complexity of the gut-lung axis in PAH-CHD. To gain a more comprehensive understanding of the relationship between microbiota and PAH-CHD, future longitudinal studies may explore how these microbiota change over time and assess their potential as predictors of the disease. After considering your

comments, we have clearly distinguished between the potential of microbiota as disease predictors and their roles in disease progression in the discussion section. The specific modifications in the revised manuscript are as follows:

“Kim et al. (2020) suggested that variations in gut microbiota may effectively predict PAH(7). Gut microbiota influence the host's immune response, metabolic processes, and resistance to pathogens through their collective metabolic activities and interactions, thereby impacting the onset and progression of diseases(8, 9).” (Page 5, lines 82-86.)

“Notably, we observed an increase in the abundance of probiotics and elevated levels of bile acids and carnitine in the microbiota of children with PAH-CHD. These observations indicate that specific microbial communities and metabolic profiles may be linked to the development of pulmonary arterial hypertension. However, this does not suggest that they can be directly utilized as predictive markers for the disease. During the mild to moderate stages of pulmonary arterial hypertension, we hypothesize that the body may initiate protective mechanisms, with these probiotics and metabolites playing a crucial role in regulating interactions between local microbiota and the host, thereby helping to prevent or slow the progression of lesions(9, 53). We believe this may reflect a natural response by the body to maintain internal homeostasis and resist disease progression, a perspective also supported by the research of Rob Knight et al(9). However, as the disease progresses, these protective factors may gradually decline. Coupled with the persistent pathological stimuli of high

pressure and high volume in the pulmonary circulation, this ultimately results in irreversible damage in pulmonary arterial hypertension. Therefore, although changes in these microbial communities during the early stages of the disease may indicate a protective adjustment, they may also represent a point of dynamic change in the disease over time. Additionally, much of Yunnan Province is located at mid-altitude, with a diverse multi-ethnic population; the geographical environment and dietary habits may also influence the microbial community structure(9).” (Pages 21-22, lines 416-436.)

Comment 3: In Figure 1B, the authors claim that the PCoA shows differences in the microbiota of the gut and lung. However, the Venn diagram shows more common species and less different ones. The different ones in the gut are all beneficial bacteria!

Response: Thank you for your valuable suggestions. We have re-examined the PCoA analysis and Venn diagram results in Figure 1B, enhancing the analysis and visualization of the relationship between grouping and species abundance. The PCoA analysis showed that the differences in microbiota composition between groups were not significant, which may relate to the stability and host dependency of core microbiota, as they play key roles in essential ecological processes such as nutrient synthesis and immune regulation in the host. Further analysis of the within-group composition ratio revealed a decrease in the Firmicutes/Bacteroidetes ratio in the gut (2.12/1.95) and an increase in the lung

(5.46/10.17), which may reflect local microbial dysbiosis, subsequently affecting immune function and metabolism. (References:1) Duan Y, *Phytomedicine*, 2024;128:155291.doi:10.1016/j.phymed.2023.155291 ; 2) Huang B, *Biomed Pharmacother*, 2023;166:115387. doi:10.1016/j.biopha.2023.115387 ; 3) Motta H , *Respir Res*,2024;25(1):211. Published 2024 May 18.doi:10.1186/s12931-024-02835-w)

Furthermore, through LEfSe analysis, we identified the specific microbial taxa responsible for these differences. Our previous description of the differential species in the gut was not sufficiently comprehensive, prompting us to make necessary revisions. We selected differential species using an LDA threshold of >3 . Among the microbial taxa enriched in the PAH-CHD group, *Bifidobacterium animalis*, *Bifidobacterium breve*, *Lactobacillus gasser*, *Clostridium butyricum*, and *Bacteroides faecis* are recognized probiotics. Their functions are consistent with high-expression metabolites in the gut and may relate to specific ecological niches within the gut microbiota, playing roles in maintaining gut barrier integrity, inhibiting pathogens, and regulating immunity. *Planctomycetota* plays a critical role in environmental carbon and nitrogen cycling and is considered a potential source of bioactive molecules, warranting further investigation. The presence of *Selenomonadaceae* in the intestines of colorectal cancer patients is associated with spleen deficiency and qi stagnation, but its specific role remains to be clarified. (References:1) Jeske O, *Front Microbiol*. 2016;7:1242.doi:10.3389/fmicb.2016.01242 ; 2) Kaboré OD, *Front*

Cell Infect Microbiol. 2020;10:519301.doi:10.3389/fcimb.2020.519301 ; 3)

Ming-Bin G, *Heliyon.* 2023;9(11):e21057.doi:10.1016/j.heliyon.2023.e21057)

Therefore, we focus on discussing the role of gut probiotics in children with PAH-CHD and plan to explore the specific mechanisms of action of these beneficial bacteria and their potential functions within the lung microbiota through animal experiments.

The specific modifications in the revised manuscript are as follows:

“Furthermore, principal coordinate analysis based on Bray-Curtis dissimilarity demonstrated no significant differences in the composition of gut and lung microbiota between groups (Figure S1B). By plotting a circos diagram to illustrate the relationship between grouping and species abundance, we found that while the dominant microbial taxa were similar between groups, their intra-group compositions differed; the *Firmicutes/Bacteroidetes* ratio in the gut decreased (2.12/1.95), while it increased in the lungs (5.46/10.17) (Figure 1B). Through LEfSe analysis, we further elucidated the differences in microbial communities between groups. In the gut, microbial communities characterized by high levels of *Bifidobacterium breve*, *Lactobacillus gasser*, and *Selenomonadaceae* underwent significant changes. However, in the PAH-CHD group, levels of *Dialister*, *Streptococcus*, and *Roseburia inulinivorans* were significantly decreased (Figure 1C). Compared to the control group, the PAH-CHD group had higher levels of *Enterococcus*, *Streptococcus peroris*, *Rothia mucilaginosa*, and *Roseburia inulinivorans* in lung microbiota, while

levels of *Thermoactinomyces*, *Leptotrichia*, *Neisseria lactamica*, and *Streptococcus parasanguinis* were lower (Figure 1D).”(Pages 6-7, lines 117-132.)

Comment 4: I did not see a strong evidence of 'gut-lung axis' involvement. KO analysis using KEGG modules to depict functional commonalities between the gut and lung microbiota is not sufficient to suggest the involvement of the gut-lung axis in PAH-CHD.

Response: Thank you for your valuable suggestions. We have re-evaluated the evidence for the involvement of the "gut-lung axis" and the validity of the KO analysis of KEGG modules. We observed significant differences in the gut and lung microbiota of children with PAH-CHD, closely associated with metabolic characteristics. Although the current commonalities in microbial species and metabolites have not fully confirmed the involvement of the gut-lung axis, we believe these changes may indirectly affect lung health through microbial displacement, metabolite expression, and immune responses, providing evidence for the role of the gut-lung axis. The KO analysis of KEGG modules revealed functional commonalities between the gut and lung microbiota, further supporting the potential role of the "gut-lung axis." This study aims to preliminarily explore the relationship between gut and lung microbiota and PAH-CHD, providing direction for future mechanistic investigations.

Furthermore, we recognize that the primary communication pathways of

the "gut-lung axis" include not only microbial displacement but also soluble microbial components, metabolites, and immune cells that connect the gut and lungs through the bloodstream and lymphatic circulation. We are conducting a multicenter clinical study to assess the systemic inflammatory status of PAH-CHD, as well as basic research to explore the anti-inflammatory effects of microbial modulation of bile acids, to further understand the role of the gut-lung axis in PAH-CHD. Based on these advancements, we plan to write a new article and hope to submit it to this journal for consideration soon.

We have also clarified the limitations of this study and future directions in the revised manuscript, as follows:

“This study identified commonalities between gut and lung microbial species and metabolites through bioinformatics analysis, but it has not adequately confirmed the role of the gut-lung axis in PAH-CHD. Therefore, future research should validate the key mechanisms underlying this hypothesis through fundamental experiments. Additionally, this study lacked an in-depth analysis of potential variables, including ethnicity, lifestyle background, and family factors, highlighting an area for further investigation.” (Pages 25-26, lines 496-502)

Comment 5: The authors speculate that microbial species may travel from the gut to the lung via blood (lines 313-315) but the metabolite analysis from blood did not corroborate with that from the gut and lung (Figure 2C).

Response: Thank you for your valuable suggestions. The manuscript mentions that the analysis of lung microbiota revealed a significant enrichment of gut symbionts *Roseburia* and *Enterococcus* in the PAH-CHD group. We explain this as "possibly due to the microbiota undergoing transient recolonization in the lungs via micro-aspiration and micro-respiration, rather than being resident in healthy individuals." In children with PAH-CHD, since the Qp/Qs ratio is greater than 1.5, the pulmonary circulation is subjected to prolonged physical stimulation due to high pressure and volume, altering the living environment of lung microbiota and creating conditions for microbial ectopic colonization. Therefore, gastrointestinal microbes are more likely to colonize the lungs through micro-aspiration from the oropharynx. The colonization of *Roseburia* and *Enterococcus* in the lungs may influence the local metabolic environment. Studies have shown that the transplantation of *Roseburia intestinalis* can increase the levels of secondary bile acids (such as CDCA) in feces and improve bile acid synthesis and transport through the bile acid/FXR-FGF15 pathway. *Enterococcus* can produce bile salt hydrolase (BSH), which deconjugates conjugated bile acids into unconjugated forms. (References:1) Sun H, *iScience*. 2023;26(12):108392. doi:10.1016/j.isci.2023.108392 ; 2) Chand D, *Biochim Biophys Acta Proteins Proteom*. 2018;1866(4):507-518. doi:10.1016/j.bbapap.2018.01.003) Therefore, we speculate that the highly expressed bile acid substances in the bronchoalveolar lavage fluid of children with PAH-CHD come from two sources: the original lung microbiota (such as

Prevotella, *Streptococcus*, *Veillonella*, *Neisseria*, etc.) and the metabolic activity of *Roseburia* and *Enterococcus*.

Certainly, the main communication pathways between the "gut-lung axis" also include soluble microbial components, metabolites, and immune cells that connect the gut and lungs through the bloodstream and lymphatic circulation. In our study, we also examined the microbiota and metabolites in the blood using 16S and metabolomics, but did not find microbes and metabolites that consistently expressed with the gut and lungs. This may be due to dilution in the blood, transport via phagocytosis by immune cells such as macrophages or dendritic cells, or epithelial barriers in the gut and lungs, which reduced their concentrations in the blood and thus did not show intergroup differences. To address this issue, future studies could explore their displaced expression using in vivo bioluminescence imaging techniques.

The following modifications and additions have been made to correct inaccuracies in the English and Chinese expressions in the manuscript.

“In children with PAH-CHD, the Qp/Qs ratio exceeds 1.5, leading to prolonged pathological stimuli from high pressure and high volume in the pulmonary circulation, which modifies the survival environment of lung microbial communities and fosters microbial ectopic colonization. Consequently, gastrointestinal microbes are more prone to colonize the lungs through oropharyngeal microaspiration.(24)” (Pages 16-17, lines 316-321)

Comment 6: Authors show that PAH-CHD samples are enriched in *Bifidobacterium animalis* which causes expression of aquaporins and tight junction proteins (lines 380-381). Did they check for gut permeability in gut epithelial cells of the patients? Are lung tight junctions affected?

Response: Thank you for your in-depth analysis and valuable suggestions. We reviewed the cited literature again and found that it did not further explore the changes in intestinal permeability of the patients' intestinal epithelial cells and their impact on lung tight junctions. Your insights encourage us to investigate the relationship between microbes and PAH-CHD from the perspective of tight junctions. Currently, we can only assess lung barrier function by measuring proteins in bronchoalveolar lavage fluid. Due to ethical constraints, it is challenging to obtain lung and intestinal tissue samples from the PAH-CHD population; however, we can explore their mechanisms of impact using experimental models.

We have refined the expressions in the revised manuscript and indicated future research directions. The specific modifications are as follows:

“The study found that *Bifidobacterium animalis subsp. lactis XLTG11* enhances the intestinal barrier by upregulating tight junction proteins (ZO-1, Claudin-1, and Occludin) in the colon, while inhibiting the TLR4/NF- κ B signaling pathway, increasing anti-inflammatory cytokines, and promoting the diversity and balance of gut microbiota, thereby strengthening host defenses(43).

Bifidobacterium breve enhances the abundance of taxa related to bile acid

regulation (such as *g_Bifidobacterium* and *g_Clostridium sensu stricto 1*), thereby promoting the uncoupling of bile acids in the intestine. In vitro cell experiments further confirmed that CA enhances the expression of tight junction proteins (Occludin and Claudin-1) and MUC2(44). In the future, we can investigate further the effects of these strains on pulmonary tight junctions and inflammatory responses, as well as how they may improve clinical outcomes in PAH-CHD patients.” (Page 20, lines 383-395)

Minor comments:

Comment 1: Line 26 states that 'Author order was determined on the basis of seniority'. Established criteria should be followed to determine the sequence of authors.

Response : Thank you for highlighting the importance of author order. We determined the author order in the manuscript based on the specific contributions of each author. We would like to correct our previous inaccurate statement:

In our study, the author order is based on the following contributions:

Jiahui Xie(First author), Conceptualization, Data curation, Formal analysis, Methodology, Visualization, Writing – original draft, Writing – review and editing ;

Xiaoyu Zhang(Co-first author), Conceptualization, Data curation, Formal analysis, Methodology, Project administration, Writing – review and editing ;

Liming Cheng(Co-first author), Conceptualization, Data curation, Formal analysis, Methodology, Writing– review and editing ;

Yao Deng, Conceptualization, Data curation, Supervision, Writing – review and editing ;

Haobo Ren, Data curation, Supervision, Validation, Writing – review and editing ;

Minghua Mu, Liang Zhao, Chunjie Mu and Jiaxiang Chen, Supervision, Validation, Writing – review and editing ;

Kai Liu(Co-corresponding author), Conceptualization, Supervision, Validation, Writing – review and editing;

Runwei Ma(Corresponding author), Conceptualization, Funding acquisition, Project administration, Supervision, Validation, Writing - review & editing.

All authors have agreed to the final order of authorship.

Comment 2: The title of the manuscript is misleading and needs correction in language.

Response : Thank you for your suggestion; we have changed the article title to “Integrated Multi-Omics Analysis of the Microbial Profile Characteristics Associated with Pulmonary Arterial Hypertension in Congenital Heart Disease”.

Comment 3 : In several places in the text, earlier works are referred to but the reference numbers are missing. Examples include line 83 (“In their 2020 study,

Kim et al highlighted---") and line 398 ("Stephen M's research indicates -----"

Response : Thank you for your reminder; we sincerely apologize for our oversight. We have re-examined the literature management and made additions, such as “In a 2020 study, Kim et al. emphasized ---” cited in reference [7]; and “Research by Stephen M shows -----,” with references [50-52] being related studies published by Stephen M. Black's team.

Comment 4 : I found most of the figures very cramped and difficult to read and understand. Many figure legends are not adequately described.

Response : Thank you for your valuable suggestion. We have reformatted Figures 1 and 3 to 5 and made corresponding adjustments to the manuscript and supplementary figures. The updated figures include Figures 1, 2, 4, and Supplementary Figures 1 and 6 to 8, which have been resubmitted for review.

Comment 5 : In some places, sentences are repeated. Example: lines 343-347 are same as lines 355-359 but the cited reference is different.

Response: Thank you for your reminder; we have carefully reviewed the manuscript. We apologize for the oversight during the editing process and have corrected the content in lines 355-359 of the original manuscript. The correct content is: “Microbiota are involved in the biotransformation and modification of bile acids, and the composition of bile acids has been shown to regulate the structure of the microbiota, which may also be one of the factors contributing

to the reconstitution of lung microbiota(37).” We have updated this description in the revised manuscript (Page 19, lines 360-363) and added the corresponding reference [37].

We tried our best to improve the manuscript and we appreciate for Editors and Reviewerswarm work earnestly, and hope that the correction will meet with approval.Once again, thank you very much for your comments and suggestions.

Re: Spectrum01808-24R1 (Integrated Multi-Omics Analysis of the Microbial Profile Characteristics Associated with Pulmonary Arterial Hypertension in Congenital Heart Disease)

Dear Dr. Runwei Ma:

Your manuscript has been accepted, and I am forwarding it to the ASM production staff for publication. Your paper will first be checked to make sure all elements meet the technical requirements. ASM staff will contact you if anything needs to be revised before copyediting and production can begin. Otherwise, you will be notified when your proofs are ready to be viewed.

Sincerely,
Benjamin Liu
Editor
Microbiology Spectrum